# Universal power law of the gravity wave manifestation in the AIM CIPS polar mesospheric cloud images

Pingping Rong[1], Jia Yue[1,2], James M. Russell III[1], David E. Siskind[3], and Cora E. Randall[4,5]

[1]Center for Atmospheric Sciences, Hampton University, Hampton, VA23668, USA

[2]Earth System Science Interdisciplinary Center, University of Maryland, College Park, MD20740, USA

[3]Space Science Division, Naval Research Laboratory, Washington DC20375, USA

[4]Laboratory for Atmospheric and Space Physics, University of Colorado Boulder, Boulder, CO80303, USA

[5]Department of Atmospheric and Oceanic Sciences, University of Colorado Boulder, Boulder, CO80309, USA

*Correspondence to*: ping-ping.rong@hamptonu.edu

**Abstract:**

We aim to extract a universal law that governs the gravity wave manifestation in the polar mesospheric clouds (PMCs). Gravity wave morphology and the clarity level of display vary throughout the wave population manifested by the PMC albedo data. Higher clarity refers to more distinct exhibition of the features which often correspond to larger variances and better organized nature. A gravity wave tracking algorithm based on the continuous Morlet wavelet transform is applied to the PMC albedo data at 83km altitude taken by the AIM Cloud Imaging and Particle Size (CIPS) instrument to obtain a large ensemble of the gravity wave detections. The horizontal wavelengths in the range of ~20-60km are the focus of the study. It shows that the albedo (wave) power statistically increases as the background gets brighter. We resample the wave detections to conform to a normal distribution to examine the wave morphology and display clarity beyond the cloud brightness impact. Sample cases are selected at the two tails and the peak of the normal distribution to represent the full set of wave detections. For these cases the albedo power spectra follow exponential decay toward smaller scales. The high albedo power category has the most rapid decay (i.e., exponent=-3.2) and corresponds to the most distinct wave display. The wave display becomes increasingly more blurry for the medium and low power categories that hold the monotonically decreasing spectral exponents of -2.9 and -2.5, respectively. The majority of waves are straight waves whose clarity levels can collapse between the different brightness levels but in the brighter background the wave signatures seem to exhibit mildly turbulent-like behavior.

## 1. Introduction

Atmospheric gravity waves play important roles in the atmospheric circulation, structure, and variability. The influence of breaking gravity waves on the dynamics and chemical composition of the 60 to 110 km region has been the most significant. The momentum deposited by breaking waves at mesospheric altitudes reverses the zonal winds, drives a strong mean meridional circulation, and produces a very cold polar summer mesopause region that enables the polar mesospheric clouds (PMCs) to form (Fritts and Alexander, 2003; Garcia and Solomon, 1985). Aside from causing these indirect but fundamental effects on the global circulation, gravity waves also have widespread displays in PMCs (e.g., Fogle

and Haurwitz, 1966; Fritts et al., 1993; Dalin et al., 2010; Taylor et al., 2011; Thuraraijah et al, 2012; Yue et al., 2014) serving as a visible manifestation of the polar summer mesospheric dynamics.

Semi-organized wave-like structures have been the most characteristic and widespread features in PMCs. PMCs are also referred to as noctilucent clouds (NLCs) when observed from the ground. In an extensive review given by Fogle and Haurwitz (1966), four types of NLCs are categorized at a descriptive level: bands and long streaks, billows, whirls, and veils. Most of these features resembled semi-organized wave signatures. These NLC features are reflections of gravity waves, gravity wave breaking, or wave breaking induced turbulence (e.g., Fritts et al., 1993; Dalin et al., 2010; Baumgarten and Fritts, 2014; Miller et al., 2015). Gravity wave horizontal scales encompass an extremely broad range of ~10-1000 km (e.g., Fritts and Alexander, 2003), but the most widespread and readily observed displays in the NLCs are at wavelengths shorter than 100 km (e.g., Fogle and Haurwitz, 1966). The near-infrared hydroxyl (OH) airglow images also revealed similar wave patterns. For example, Taylor and Edwards (1991) observed several ~15-20km wavelength linear wave patterns over Hawaii in March, and Yue et al. (2009) reported capturing the mesospheric concentric waves with wavelengths in the range of ~40-80km over Colorado and a few neighboring states.

The Aeronomy of Ice in the Mesosphere (AIM) satellite was launched in April 2007 becoming the first satellite mission dedicated to the study of PMCs (Russell et al., 2009). One of the primary research goals of the AIM mission is to explore how PMCs form and vary. In pursuing this goal, gravity waves have become an increasingly important topic in AIM science investigations. The Cloud Imaging and Particle Size (CIPS) instrument (McClintock et al., 2009) aboard the AIM satellite provides PMC images that cover the polar region daily throughout the summer season in both hemispheres, and has collected almost 10 years of data to date. These data have enabled extensive studies of gravity wave signatures in PMCs and of mesospheric dynamics more generally (e.g., Thurairajah et al., 2013; Yue et al., 2014). Thurairajah et al. (2013) presented a host of characteristic cloud structures in the CIPS PMC images, among which the noteworthy ones include the "void" feature with a clean edge and a core region of sharply reduced cloud brightness, and concentric waves (see also Taylor et al., 2011; Yue et al., 2014). These are fairly unique signatures with low occurrence frequency and are not the focus of the current study although we do also discuss some examples of the concentric waves in a later part of this paper. Yue et al. (2014) correlated concentric wave patterns in the CIPS PMC data with similar patterns in the stratosphere observed by the Atmospheric Infrared Sounder (AIRS) instrument aboard NASA's Aqua satellite (Aumann et al., 2003). Concentric waves are the most evident proof that gravity waves excited by tropospheric storm systems have propagated into the mesosphere.

Quantitatively characterizing gravity waves in PMC images is generally difficult because the wave patterns are complex although several NLC morphology types have been successfully interpreted in the previous modeling studies (e.g., Fritts et al., 1993; Baumgarten and Fritts, 2014). In addition, PMCs are characterized by a hierarchy of larger scale features (~100-1000 km) that often obscure the smaller scale (<100 km) gravity wave signatures. It is worth noting that these large scale features may not be exclusively gravity wave structures because tides and planetary waves also play important roles in modulating the PMC spatial variability [e.g., Merkel et al., 2009].

In this study we developed an algorithm to quantify the occurrence and manifestation of gravity waves with horizontal wavelengths of ~20-60 km in the CIPS level-2 albedo data (Lumpe et al., 2013). Such a scale range is chosen because the full display of these waves can well fit into one CIPS orbital strip and also because these short wavelength waves are proven to be the most commonly observed. We aim at obtaining a universal law that governs the wave display for a large ensemble of semi-organized wave structures via sorting their albedo disturbance power and examining their relationship with the background cloud brightness. The law obtained will go beyond the apparent dependence of the albedo wave power on the background cloud brightness because the real effect of the gravity waves on the PMC brightness is through dynamical or microphysical control that are related to the variability in winds, temperature, and water vapor, as is shown in the model studies carried out by Jensen and Thomas (1994) and Chandran et al. (2012) for instance. These studies suggested that both long and short gravity waves eventually reduce the cloud brightness locally. Generally speaking, the fundamental relationship between the gravity waves and the PMC brightness is not yet fully understood. In this study we do not pursue such fundamental relationship or attempt to characterize specific wave events in a strict sense.

The algorithm we designed for CIPS differs from the wave tracking approaches proposed in some other research papers (e.g., Chandran et al., 2010; Gong et al., 2015) in the sense that it confines the scale range first, rather than searching for spectral peaks. This is because the wave patterns in PMCs can be obscured by larger scale variability that possesses larger amplitudes and also because these patterns are rarely monochromatic. As a result, spectral peaks, for example in wave numbers, do not stand out easily unless a distinct wave structure was first visually detected and then a spectral analysis is carried out along its most optimum orientation. Such a challenge is also reflected in the airglow image processing that aims at identifying the mesospheric gravity waves (e.g., Matsuda et al., 2014). In addition, it is worth mentioning that in the current study we did not adopt high-pass filtering to extract the small scale structures (e.g., Chandran et al., 2010) because 2-dimensional (2-D) filtering is prone to inducing notable artificial features if large and small scales are not separated optimally. Another important technique to detect gravity waves and to resolve their characteristics is via applying a spatio-temporal analysis to either the ground based or satellite measurements (e.g., Wachter et al., 2015; Ern et al., 2011). For example, Wachter et al. (2015) applied such a technique to the OH airglow time series measured at the chosen triangular equilateral ground sites to yield a consistent set of wave parameters. In these analyses however a full display of the waves is not captured because only a few locations are used. CIPS on the other hand has extended spatial coverage and therefore in the current study we aim to only detect the spatial wave patterns.

The planning of the paper is as follows. A brief description of the CIPS data is given in Sect. 2. In Sect. 3 we describe the wave tracking approach, provide an analytical demonstration, and then apply the algorithm to a few concentric wave patterns found in the CIPS imagery. In Sect. 4 the statistics of the wave power is obtained and its dependence on the background cloud brightness is quantified. A resampling process is applied to reach a normal distribution of all the wave power values. Resampling serves as the first step to further examine the gravity wave display beyond the apparent impact of the cloud brightness. In Sect. 5 representative cases are chosen to extract a universal law that controls the wave display. As a step further, demonstrations of these cases are shown to verify. Conclusions are given in Sect. 6.

## 2. CIPS dataset

CIPS version 4.20 level-2 orbital strips of PMC albedo at 83km altitude are used in this study (Lumpe et al., 2013). CIPS is one of the two instruments that are currently operating aboard the AIM satellite. The CIPS instrument (McClintock et al., 2009) is a panoramic imager viewing nadir and off-nadir directions to measure ultraviolet radiation (centered at 265nm) scattered by the clouds and atmosphere. In the spectral region near 260 nm, absorption of ozone in the lower atmosphere renders the earth nearly dark, maximizing the contrast of PMC scattering relative to the atmospheric background. It is worth mentioning that aside from the PMC data used in this study, the CIPS Rayleigh Albedo Anomaly (RAA) data are also made available to the public. The RAA data characterize the gravity waves at altitudes of 50-55km (Randall et al., 2017) serving as the only existing imaging dataset that can reveal the gravity wave horizontal structures near the stratopause. CIPS consists of four wide angle cameras arranged in a "bowtie" shape that covers a 120° (along orbit track) × 80° (cross orbit track) field of view (FOV). It measures scattered radiances from PMCs near 83km altitude to eventually derive cloud morphology and particle size information. Each individual cloud had a stack of maximum seven exposures from different view angles from which we derive PMC scattering phase functions and eventually retrieve the nadir horizontal spatial features. CIPS horizontal resolution is approximately 2-7 km depending on how far a given pixel is from the center of "bowtie", with the center of the "bowtie" possessing the finest resolution. CIPS version 4.20 retrieval algorithms and data products are described in detail by Lumpe et al. (2013). In level-1a the camera flat-fielding is applied to remove the pixel-to-pixel variation induced by each camera, and then normalization between the cameras is applied. In level-1b all cameras are merged to create a consistent set of CIPS measurements. In this stage the measurements are adjusted onto a common grids system of 25km$^2$ resolution. In the level-2 processing the cloud scattering signal is further distinguished from the background Rayleigh signal based on their different scattering angle dependence. The level-2 retrieval is operated on the level-1b data, so that the level-2 data product is registered on the same 25km$^2$ resolution grids. Throughout the 10 years of AIM mission the CIPS retrieval has experienced earlier versions and the theoretical framework of these retrievals was described in Bailey et al. [2009]. In the previous CIPS data versions the Rayleigh background was retrieved pixel-by-pixel rather than over the entire orbital strip like in version 4.20. This earlier approach will result in increased retrieval noise in the cloud parameters, requiring additional smoothing procedure to increase signal-to-noise at the expense of retrieval resolution.

## 3. Wave tracking algorithm

### 3.1. Analytical demonstration

One-dimensional (1-D) continuous wavelet transform (CWT) calculations constitute the basic elements of the proposed CIPS wave tracking algorithm. The term "wave tracking" in the context of this paper refers to the operation of tracking all existing quasi-periodic wave displays in the CIPS PMCs over the scale range of ~20-60km rather than tracking a specific wave event. We first demonstrate the effectiveness of the approach using an analytically composed series. When the algorithm is applied to CIPS (see Sect. 3.2 and later), each individual CWT calculation will be carried out along a presumably ~400km, or 80 grids length of CIPS PMC albedo segment, and will deliver a total number of 22 components that spans the scales 2.0-76.0 in grids (5km/per grid). The relevant scale range for this study is ~4.0-12.0 grids (~20-60km), and

the total power of the relevant CWT components is termed as CWT power, or albedo power in the following CIPS related discussion. It is worth pointing out that a ~20-60km scale range is focused in this study because their total albedo power spatial distribution corresponds well with the readily observed wave signatures in the albedo maps (see the following Fig. 2, and Figs. 7-9 and 11). The 60km threshold appears particular but it is chosen simply because it is among the sequence of individual scales for the CWT calculations, which are 4.0, 4.7, 5.6, 6.7, 9.5, and 11.3 grid units respectively. A radius of roughly ~400km is chosen because it is able to include many repeats (~10) of the wave ridge and trough to provide the full extent of wave display. Yet the spatial span of the wave display should not be overly extended because we do not wish to go across several wave events or different types of variability along the path of CWT calculation. These calculations will be carried out along all 360° radial directions (3° increment) when being applied to CIPS PMCs.

A 6$^{th}$ order Morlet wavelet is adopted in this study. Morlet wavelet (Gabor, 1946) is a complex exponential (plane sinusoidal waves) windowed by Gaussian so that both periodicity and localization can be realized, defined as $e^{ikx/s}e^{-x^2/(2s^2)}$, where $k$ is the (non-dimensional) order and $s$ is the scale. Scale $s$ determines both the width of the Gaussian and the period of the sinusoidal signal. In a 6$^{th}$-order Morlet wavelet ($k=6$) the scale $s$ is almost precisely the period of the sinusoidal signal. We emphasize here that in the following main analysis applied to the CIPS PMC images the scales of the wave structures refer to the Morlet wavelet scales. However, we will first use an artificially created series to demonstrate that CWT and Fast Fourier Transform (FFT) deliver qualitatively consistent results.

The artificially created series, which is shown as the red thick curve in Fig. 1a, is composed of seven plane sinusoidal waves that have the wavelengths from 4.0 to 12.0 grids and zero initial phases at the start point of the series. These wavelengths are the 5$^{th}$-11$^{th}$ scales delivered in a CWT calculation mentioned above. The shortest and longest of these plane waves are shown by the gray and green dashed lines in Fig. 1a. The total CWT power over the scales of 4.0-12.0 grids is shown as the black curve in Fig. 1b, whereas the red thick curve in Fig. 1b is the reconstructed series using the Morlet wavelets and the corresponding CWT coefficients. It is notable that the black curve is smooth and exactly follows the magnitude change of the localized shorter scale signals. The basis vectors of CWT are not orthogonal and therefore a reverse CWT does not exist in a strict sense, but we do find that the reconstructed series greatly resemble the original series although their magnitudes slightly differ. This suggests that the CWT and reverse CWT work efficiently on a quasi-periodic signal series. Especially, the fact that CWT almost precisely captures the local variability of the wave amplitude suggests that the CWT algorithm will be an effective approach to detect the gravity waves in the CIPS PMCs.

Fig. 1c shows the CWT spectrum of the created series. We just mentioned that the created series is the sum of only seven FFT components, but since FFT and CWT have different basis vectors the CWT will project onto all the scales from 2.0 to 76.0 grids. Over the seven relevant scales from 4.0 grids to 12 grids the slope on the double-logarithm diagram is as weak as -0.54 reflecting the fact that we have adopted identical amplitudes for all the plane sinusoidal waves used to create the series. For scales < 4.0 grids there is an extremely rapid decrease of the CWT power density with a slope of -12, indicating that these scales are almost non-existent. It is noteworthy that the created signal exhibits quasi-periodic nature even though there are no spectral peaks. However, due to the involvement of multiple scales, the fluctuation washes out at

certain portion of the series (i.e., in the range of 3.0-23.0). In this latter case, quasi-periodicity is impaired by the involvement of multiple components.

## 3.2. Demonstrations using the concentric wave patterns

In terms of applying the algorithm to the CIPS PMC images, a direct 2-D CWT routine would be preferred but does not exist in the standard numerical recipe. In addition, there is an ambiguity on determining the phases in the CWT algorithm because there is always a tradeoff between the localization of the signal and a clear phase determination.

In this study we used an algorithm based on expediency as well as efficiency consideration. The 1-D CWT calculations are carried out along the full 360° radial directions (3° increment) centered at a given location within the CIPS albedo orbital strip. The resampling of the CIPS data is performed on the radial and angular directions centered at such a location. Along the radial direction the step of increment is 5 km which is the same as the CIPS level-2 resolution, while in the angular direction a step of 3° is chosen based on the consideration of yielding a sufficiently detailed yet smooth spatial map of the albedo wavelet power. This approach was inspired by the intent of detecting ideal concentric waves because such a design will result in the maximum CWT albedo power within a given radius. In addition, performing the CWT along all radial directions will efficiently capture the waves of all orientations. Since the basis vectors of 2-D FFT are straight (linear) waves of all orientations, the current algorithm is more or less a short version of the 2-D FFT in a localized area but has the merit of being straightforward in reflecting the local albedo wave power.

The specifics of the algorithm are described as follows. An elliptical region of 80 grids along orbit and 80×0.65 grids cross-orbit is used to carry out the CWT calculations. The factor 0.65 is empirically chosen based on a few fitted examples of the concentric waves found in CIPS. The rationale of such a factor will be revisited in a following paragraph. Concentric waves found in CIPS appeared to be mostly elongated in the along-orbit direction. This factor has no qualitative effect on the results except that when the elliptical region fits an actually existing concentric wave pattern, it achieves the largest albedo CWT power, which is a desired condition for the wave tracking operation. The elliptical region is moved around to fully cover the orbital strips. The steps of the movement are the half axial lengths in both along-orbit and cross-orbit directions. Except for the intent to achieve the full coverage of the orbital strip, the "move-around" scheme will also ensure the capture of the albedo CWT power from varying orientations in the same region. The albedo CWT power map enclosed in the given elliptical region measures the total albedo fluctuation intensity in the scale range of ~20-60km for each spatial location.

Five examples of concentric waves and the corresponding albedo power maps are presented in Fig. 2 to demonstrate how the wave tracking algorithm works for CIPS. Concentric waves are chosen because they are well documented as possessing a unique morphology and meanwhile serving as a proof of the connection between the lower and higher atmosphere. These waves are extremely rare and were detected only a few times for a given PMC season. The percentage of detection is less than 5% in terms of days per season, and by spatial coverage the fraction is even smaller. All examples have shown only partial rings. The model results by Vadas et al. (2009) simulating the concentric rings to compare with the observations near Fort Collins Colorado have shown that including realistic zonal winds can substantially disrupt the

completeness of rings in the mesosphere, with about 50% of the wave structure being disrupted. In the CIPS PMCs the rings are more severely disrupted. In some cases only a 20 degree section of the full 360 degree circle has survived (not shown). The model results by Vadas et al. (2009) also indicate that if a July zonal wind is adopted in the simulation the rings will be elongated to an axial ratio of about 0.6-0.7, which roughly agrees with the findings in CIPS. It appears that in summer the concentric waves will likely be substantially elongated compared to those in winter or spring.

The example shown in Fig. 2a has a wavelength of about 60 - 80km which is longer than most other concentric waves in the CIPS PMCs, but since the (bright) ridges are much narrower than the (dim) troughs the albedo power still reaches notable magnitudes. Remember that in this study only the CWT power within the wavelength range of ~20-60km is calculated, and what is shown in Fig. 2a again reminds us that wave patterns in PMCs only achieve quasi-periodicity. The example in Fig. 2b is a set of highly distinct concentric waves. The albedo power reaches notably larger values when the waves are the most distinct in the upper-right quadrant of the albedo map (see Fig. 2b upper panel). In the upper-left quadrant of the albedo map the waves become blurry but one can still tell that they are concentric. In the blurry part of the map the albedo power decreases sharply, as is clearly seen from the corresponding lower panel. Given the confined scale range of ~20-60km, being visually blurry can be interpreted as possessing lower variance level which may have been a result of diffusive processes. In the spectral space, if the leading scales stand out poorly from the neighboring smaller scales then the structure will be less organized which also makes it less distinct. This topic will be further addressed in the following Sect. 5. Within the same wave display, varying from being distinct to blurry must be controlled by some physical process yet to be unraveled. In the lower-right quadrant of the albedo map there are some straight waves that have cross-interfered with each other but are much less distinct than the main part of the concentric waves. This suggests that multiple wave packets of different morphology often coexist right next to each other in the PMCs. Fig. 2c shows an example of extremely faint concentric waves which are characterized by much lower albedo power than those in the rest of the examples. Fig. 2d shows a slightly weaker but also fairly distinct partial concentric wave pattern that also coexists with some blurry straight wave patterns. Fig. 2e is an example of concentric wave in a brighter background that occurred in 2009. In 2009 CIPS orbits appear twisted due to the turning of the camera pointing.

At the end we must also point out an inherent drawback of the CWT wave tracking algorithm. It is apparent that in all the examples shown in Fig. 2 the clouds (e.g., $>5.0\times10^{-6}$ sr$^{-1}$) do not fill up the entire elliptical region and furthermore the wave signatures are mostly partial rings. If such an inhomogeneity is strong, the mean albedo power or background brightness can misrepresent the characteristics of the region. We therefore have chosen a relatively small radius (~400 km) to carry out the calculations to minimize the effect of inhomogeneity. Due to the high complexity of the PMC signatures it is unlikely to simply eliminate such an effect. Nevertheless, we so far did not run into any noteworthy problem due to such effect when analyzing the wave tracking results.

## 4. Statistics of the gravity wave albedo power values

### 4.1. Brighter PMC background threshold

The statistical ensemble consists of the albedo power values averaged within all elliptical regions used to carry out the wave tracking. We need to emphasize here that the albedo power refers to the total power within the scale range of ~20-60km. The wave tracking procedure has been carried out throughout the two northern summers in 2007 and 2010 from June $1^{st}$ to August $31^{st}$. We take a particular interest in the wave display in the brighter PMCs because the previously identified waves are mostly residing in the relatively dim cloud environment. We split the cloud population into two subsets, one with only 0-2% of bright clouds (with a threshold of $25\times10^{-6}sr^{-1}$) in the elliptical region, which is the overall dimmer cloud group, and the remaining set contains systematically larger fraction of bright clouds. The $25\times10^{-6}sr^{-1}$ threshold is chosen empirically. The bright cloud presence frequency (denoted by $freq_{25}$ hereinafter) is a better index than the mean cloud albedo in characterizing a systematically brighter cloud background.

## 4.2. Histograms of the albedo power values

Fig. 3 shows the histograms of the albedo power values for the full set of wave detections (in red) and those residing in the brighter background (in black), and both show a peak number density being close to zero and then a rapid decrease as the albedo power increases. Peak locations approaching zero indicate that the majority of the waves are in the range of low albedo power. It is especially worth noting that the full set and the brighter set collapse as the albedo power is greater than $\sim20\times10^{-12}sr^{-2}$ whereas before this point they separate substantially. This indicates that brighter clouds mostly coincide with the higher albedo power and the waves that have caused the difference in the two curves reside in the dimmer group. In addition, the collapsing part of the curves forms a straight line under a logarithm vertical axis, suggesting an exponential decrease of the wave detection number density as the albedo power exceeds the value where the peak number density occurs. For the full set of the wave detections the straight line proceeds to much smaller albedo power value while for the brighter set it shows a peak at a higher albedo power. Removal of the dimmer cloud group by a given threshold caused this. In general, the exponential decrease of the wave detection number density toward increasingly higher albedo power is a robust result but how rapidly the number density decreases may show inter-annual variability (not shown). It is worth mentioning that the analysis of the PMC ice water content measured by the Solar Backscatter Ultraviolet (SBUV) instruments (DeLand and Thomas, 2015) yielded the same type of distribution. These authors further investigated the apparent inter-annual variability of the distribution slope and concluded that population ratios between the hierarchy of particle sizes may have been different for individual years to cause this variability. Later in this paper we will find that PMC albedo and the corresponding wave power hold a statistically linear relationship and therefore it is within expectation that both the PMC intensity and the wave power follow similar distributions. Fig. 3 also shows that when the albedo power reaches its upper limit the curve flattens out, but such a behavior is simply caused by the low sample number which is of no significance.

## 4.3. Relationship between the albedo power and $freq_{25}$

Although the dimmer subset takes up a major fraction of the cloud population, which exceeds 65%, we take just as much interest in the wave display in the brighter cloud background in this study. Fig. 4 shows the scatter plot of the wave detections in the brighter cloud background on the plane of albedo power versus $freq_{25}$. In Fig. 4a, within equally spaced bins

of freq$_{25}$ and the albedo power, the rainbow colored squares represent the number density of the wave detections. The dimmer group of PMCs are collapsed into the first bin of freq$_{25}$=0-0.02 and is not shown in this scatter plot to avoid any discontinuity induced by the artificially chosen threshold (i.e., $25\times10^{-6}\mathrm{sr}^{-1}$).

Fig. 4a shows that the wave detections (see rainbow colors) are grouped more densely in the low albedo power as well as the low brightness region. This generally agrees with what Fig. 3 just showed but we should note that the wording "low" is in a relative sense because we have taken away the dimmer set in this analysis. For any given freq$_{25}$ the wave detection number density distribution resembles a normal distribution but the outliers are strongly asymmetric showing much further-reaching albedo power value at the upper limits. In addition, we find that as freq$_{25}$ increases the peak number density moves toward an increasingly larger albedo power, shown by the black dashed curve. Both suggest that in a statistical sense larger albedo power corresponds to brighter cloud background. At the dimmer end where freq$_{25}$=0.02 the albedo power is within $50\times10^{-12}\mathrm{sr}^{-2}$ (amplitude $\sim7.0\times10^{-6}\mathrm{sr}^{-1}$) while for freq$_{25}$=1.0 it reaches values greater than $150\times10^{-12}\mathrm{sr}^{-2}$ (corresponding to an amplitude of $\sim12.0\times10^{-6}\mathrm{sr}^{-1}$).

We next adopt an analytic form to parameterize the relationship between the albedo power and freq$_{25}$ in order to resample the wave detections into a consistent normal distribution. This is a preliminary step taken for the future removal of the apparent dependence of the wave power on the background cloud brightness. The mechanism that controls such an apparent dependence is not pursued in this paper and a future study will be required to understand this since we have learned that the previous modeling studies do not seem to directly interpret it. For example, Chandran et al. [2012] have shown that both the short-period and long-period gravity waves ultimately reduce the domain averaged PMC brightness.

A set of square root sectioning curves (albedo power=factor$\times\sqrt{\text{freq}_{25}}$) are used to split the wave detections into subsets to achieve the normal distribution. Such an analytic form is chosen because it roughly coincides with the black dashed curve in Fig. 4a that shows how peak number density of wave detections varies with freq$_{25}$. The interval of the sectioning curves is by factor=$1/2^{i/4}$ with index $i$ varying from 0 to 21, shown in Fig. 4a. Totally twenty-three sections are used to produce a smooth probability distribution, with the first index i=-1 being the closest to the albedo power axis. The $2^{i/4}$ rather than a linear sequencing is chosen to account for the fact that the data points become increasingly denser from being close to the albedo power axis toward the freq$_{25}$ axis. Fig. 4b confirms that the resampling produces almost a precise normal distribution, and at the 11$^{\text{th}}$ interval it reaches the peak which splits the wave detections in half.

**4.4. Relationship between the albedo power and mean albedo**

In this subsection we reexamine the albedo power dependence on the background cloud brightness using the mean albedo within the elliptical region as the horizontal axis. This angle of investigation provides a smoother picture because it does not use any imposed threshold ($25\times10^{-6}\mathrm{sr}^{-1}$). Two purposes are served by doing so. First, we tend to include the wave detections with dimmer background since they take a major fraction of the cloud population as is mentioned above. Second, we tend to examine whether the full set of wave detections and those residing in the brighter cloud background follow a consistent statistical relationship between the albedo power and the background cloud brightness.

Fig. 5a shows a similar scatter plot except using the mean albedo (within the elliptical region) as the horizontal axis. The isopleths of the wave detection number density suggest a linear relationship between the albedo (fluctuation) power and the background albedo and we also note a strong asymmetry of the albedo power distribution between the lower and upper limits suggesting an apparent dependence of the albedo power on the background mean albedo. This confirms that albedo power monotonically increases with the background cloud brightness except that a different analytic form will be used to carry out the sectioning procedure.

Linear sectioning lines are applied to the plane of the albedo power versus the mean background albedo to yield a normal distribution. The sectioning lines are emitted from (0, 0) point and the angular interval gradually increases from the lower-right to upper-left corner to achieve symmetry of the distribution. Totally 27 intervals are used. Such a sectioning and resample procedure makes both the full set and the dimmer subset achieve the normal distribution, shown in Fig. 5b, which confirms a consistent behavior between the dimmer set and the full set. The peak of the normal distribution is reached at the 12[th] sectioning interval. Both the current Fig. 5 and the previous Fig. 3 (i.e., the albedo power histograms) indicate that dimmer subset and the full set follow a similar behavior.

### 4.5. Representative cases on the scatter plot

The dots of three different colors in Fig. 4a are the sample selections of the wave detections for future demonstration purpose. Wave tracking yields a large number of detections and we aim at obtaining a universal law that governs the full set of wave display. The selections are made roughly at the two tails, and the peak of the normal distribution (Fig. 4b), labeled as the high, medium, and low albedo power categories. Note that these categories are chosen based on the combination of both albedo power and the background cloud brightness. Three brightness levels, $freq_{25}=0$, 0.4, and 0.8, are used and therefore totally nine selections are made. The cases at $freq_{25}=0.4$ and 0.8 almost precisely follow the sectioning curves. While at $freq_{25}=0$ the three selections are made based on a linear relationship with the two selections at $freq_{25}=0.4$ and 0.8 respectively. This is because the dimmer subset ($freq_{25}<0.02$) is not included in Fig. 4a. We should point out that the sectioning curves serve only as the guidance to select the cases that reasonably cover the full set of detections but our conclusions are not sensitive to what exact cases are chosen.

We place the same nine selected cases in Fig. 5a and find that the three albedo power categories also roughly follow the linear sectioning lines and are also located approximately at the lower and higher tails, and the peak of the normal distribution.

At last we check on where the concentric waves shown in Fig. 2 turn out on the scatter plots. From looking at the turquoise colored triangles we find that they do not seem to preferably occur at any specific combinations of albedo power and background brightness. Rather, the five cases approximately evenly spread over the core region, and none has occurred in either high albedo power or high background brightness ranges. Also, it is worth mentioning that the case of the most distinct concentric waves shown in Fig. 2b has a combination of $freq_{25}=0.016$ and albedo power$=25.4\times10^{-12}sr^{-2}$, suggesting that the combination of relatively low background brightness and relatively high albedo power seem to correspond to the best clarity of wave display. Although rare in occurrence and possessing a known particular form of driving mechanism,

concentric waves seem to have shown a regular behavior in terms of the correspondence between the albedo power and the background cloud brightness.

## 5. Wave displays of the representative cases

### 5.1. Albedo power spectra

The albedo power spectra of the nine selected detections are shown in Fig. 6 with the three brightness levels being collapsed to each other for each albedo power category. These power spectra are calculated to extract a universal law that governs the wave display. The power spectra of all orientations (full 360° with 3° step) within a given elliptical region are averaged for each individual scale in the range of ~20-60km. Although inhomogeneity exists between different orientations, the spectra of the main wave signatures will dominate the mean spectrum. Unlike the FFT power spectra that are prone to exhibiting spikes (not shown) the CWT power spectra have blunter features. Especially, due to the spatial average it is even less likely that spectral peaks will stand out unless a given scale within the ~20-60km range is consistently dominant throughout the gravity wave population, which is not the case.

Based on the nine representative cases, the general form of power spectra can be expressed as $A \times (1/\text{wavelength})^{\alpha}$, where coefficient $A=1.42 \times 10^{-4}$, $1.45 \times 10^{-4}$, and $1.49 \times 10^{-4}$, and the spectral exponent $\alpha = -2.5$, $-2.9$, and $-3.2$ for the low, medium, and high power categories respectively. Note that the three categories are defined in the previous Sect. 4.5. For a confined range of wavelengths, i.e., ~20-60km, the higher wave power or larger variance level of the display will correspond to higher clarity of wave display, or sharper features. More rapid decay toward the smaller scales will also contribute to higher display clarity because the leading scale will be more dominant over the smaller scales and therefore the wave signature will be better organized. The actual wave displays shown in the following Sects. 5.2 and 5.3 will verify these arguments.

The black lines in Fig. 6 are for the high power category (see the black dots in Fig. 4a). The three brightness levels possess a consistent exponent $\alpha=-3.2$ and therefore via simple adjustment of A (in the analytic form) a normalization or collapse between the three spectra is achieved. The factors of normalization toward the brightest level are defined as follows:

$$Ratio_{0.4/0.8} = albedo\ power_{at\ freq_{25}=0.4} / albedo\ power_{at\ freq_{25}=0.8}\ , \tag{1}$$

$$Ratio_{0.0/0.8} = albedo\ power_{at\ freq_{25}=0.0} / albedo\ power_{at\ freq_{25}=0.8}\ , \tag{2}$$

where albedo power refers to the total wave power over the scales of ~20-60km. As is argued above, the normalization is carried out to examine the wave morphology beyond the wave power dependence on the background brightness. After applying these factors the three brightness levels will achieve a nearly fully collapsed power spectrum and exhibit the same level of display clarity (see Sect. 5.2). The black line under the linear axes shown in Fig. 6b indicates that the high power category possesses both the highest overall power level and the most rapid decay rate toward the smaller scales.

The error bars in Fig. 6 are the uncertainty ranges over the three brightness levels for the different scales and the exact match occurs at the middle data point (i.e., ~33km of wavelength or 0.03 of spatial frequency). It is noteworthy that both the high and medium albedo power categories have very small error bars suggesting a strong collapse of spectra at the

three brightness levels. The low albedo power category however has notably larger error bars. This means that when the albedo power reaches very low values the exponent α maintains poorer consistency between the three brightness levels. Measurement noise may play a role in causing this. For waves of small amplitudes and especially with very dim background, the noise will be strong enough to affect the determination of the wave amplitude.

### 5.2. Wave displays for the different albedo power categories

Before examining the wave displays we first set the rules of presentation. First, a white-blue color scheme with linear Red-Green-Blue (RGB) code system is used to generate the color bars. Second, the mean albedo within the elliptical region, or the background cloud brightness, is subtracted. Third, the maximum and minimum albedo deviation is set to be $\pm 20.0 \times 10^{-6} sr^{-1}$ for $freq_{25}=0.8$, and is then reduced by factors $\sqrt{Ratio_{0.4/0.8}}$ and $\sqrt{Ratio_{0.0/0.8}}$ for $freq_{25}=0.4$ and $freq_{25}=0.0$ respectively. As is argued above these factors are expected to unify the display clarity between different brightness levels.

### 5.2.1. The high albedo power category

Fig. 7 presents the CIPS albedo maps and the albedo power maps within the elliptical region for the high albedo power category (corresponding to the black dots in Fig. 4a). We observe generally wide spread and distinctly clear semi-organized structures at all three brightness levels. The three panels show very similar levels of display clarity because their albedo power spectra are almost fully collapsed with the factors $\sqrt{Ratio_{0.4/0.8}}$ and $\sqrt{Ratio_{0.0/0.8}}$ being applied. The high power category obviously possesses the overall highest power level, and furthermore it also has the most rapid albedo power decay rate (Fig. 6b) toward the smaller scales, i.e., with $A=1.42 \times 10^{-4}$ and $\alpha=-3.2$ respectively. Both conditions contribute to the fact that these displays are the most distinct among the three categories. In terms of morphology, the wave signatures resembled straight waves or interference of the straight waves but occasionally the straight wave features show curvatures at certain portion of the display (in Figs. 7b and 7c). Overall speaking the wave morphology is qualitatively consistent regardless of the brightness levels. However, looking more closely we do perceive a minor difference between the low and high brightness display. That is, at the dimmest level ($freq_{25}=0.0$) the wave display appears to exhibit stronger linearity than those in the brighter backgrounds. From the previous Fig. 2b we did notice that a dimmer background has supported highly distinct wave structures that resembled linear waves. On the contrary, the displays at $freq_{25}=0.4$ and $freq_{25}=0.8$ seem mildly turbulent-like.

### 5.2.2. The medium albedo power category

Fig. 8 presents the maps for the medium power category (corresponding to the yellow dots in Fig. 4a). Remember that these cases are the closest to the peak of the normal distribution and therefore is the most representative to the full set of wave detections. The maps shown in Fig. 8 are significantly more blurry than those in Fig. 7 but the wave signatures remain well organized. In the low brightness end ($freq_{25}=0.0$) there are interfering straight waves approximately oriented perpendicularly to each other. At $freq_{25}=0.4$ the features resembled one-directional straight wave signatures. In the high brightness end ($freq_{25}=0.8$) there are knot-like signatures which are apparently deviated from the typical linear wave signatures. Again it is worth mentioning that the three brightness levels are considered having the same level of display clarity due to the collapse of their CWT power spectra.

### 5.2.3. The low albedo power category

Wave displays at the low albedo power category (see Fig. 9) provide firm proof that the display clarity of the wave signatures becomes increasingly poorer as the albedo power decreases. All panels show highly blurry features, and yet the orientations of the wave signatures remain recognizable. Note that in this category normalizing the clarity level between different brightness levels has run into larger uncertainty because of the larger error bars (see Fig. 6a). It is noticeable that the wave display for $freq_{25}=0$ seems the most blurry because under this condition both the background mean albedo and the albedo disturbances are likely strongly affected by the measurement noise. Although display clarity does not hold any absolute physical meaning, we can conclude that the gravity wave signatures of ~20-60km wavelength become increasingly more diffusive as the corresponding albedo power decreases. Note that in the previous Fig. 2b it shows that even within the same elliptical region (~400km range) the wave display clarity differs drastically. This could be due to the unknown local forcing mechanisms that have exerted different levels of diffusion on the PMC albedo structures. So far we have come to an understanding that semi-organized structures seem extremely wide spread with a hierarchy of different albedo power levels. This is against the belief that a wave detection procedure should yield unequivocal results.

### 5.3. Artificially raise the medium toward high albedo power

The three lines with different colors in the above Fig. 6a have a hierarchy of albedo power levels as well as different slopes characterized by varying A and α. They however appear parallel to each other because the standard deviation of the exponent α (-3.2 to -2.5) only reaches 0.3. We next systematically raise the medium power level to match the high power level by increasing the coefficient A but maintain the exponent α to check on how the display clarity improves. We have argued that parameters A and α both have a control over the display clarity. Carrying out this experiment is to test the role of the coefficient A in determining the display clarity. The previously determined A for the medium power category is $1.45 \times 10^{-4}$ and in this experiment a new parameter $A_{new}=2.778 \times A$ is used to draw closer the spectra of the medium toward the high power levels. The factor 2.778 is chosen to maximally match the high and medium power spectra over the scales of ~20-60km.

The result of the amplified case is shown in Fig. 10 and we used the case shown in Fig. 8a to conduct this experiment. Fig. 10a shows the albedo map and the power map for the amplified case, and Figs. 10b and 10c are repeats of Figs. 8a and 7a, to make comparisons. It shows that the Fig. 10a exhibits notably improved clarity compared to its previous version, and as a result Figs. 10a and 10c show clarity levels drawn closer but the amplified case is still more blurry. This indicates that the difference between the two exponents (-3.2 versus -2.9) has a fundamental effect on the wave display clarity. At the end we need to point out that it is not straightforward why exponent α varies between different albedo power levels and why it remains roughly consistent along the sectioning curves used to resample the wave detections. The physical mechanism that governs such variability is worth a further investigation. It is probable, as we have argued above, that noise contamination is one cause of the smaller slopes for lower albedo power.

### 5.4. Explore the longer wavelength wave display

Longer wave length (>100km) waves are not as visually detectable as the shorter waves because the CIPS orbital strip is not able to embrace many repeats of ridge and trough of such waves. Chandran et al. (2010) used a wave detection

algorithm to yield a peak population of waves at scales of ~250 km from the cross-orbit traces. Zhao et al. (2015) yields a peak wavelength at ~400km from the along-orbit traces. Different traces may be the cause of the different peak wavelengths. Based on these studies, gravity waves of all scales are likely widespread in the PMCs. The waves at <100km wavelengths are more visually detectable because their full displays are well captured. But these small-scale waves possess a lower variance level than the larger scale waves. Using a spectral analysis the larger scales often stand out as the dominant wave events. In addition, wave event counting is not a deterministic procedure. For example, in the current study, the wave detections are forcibly confined within the elliptical regions. Generally speaking we have to cope with a lot of challenge and uncertainty in the wave tracking study.

Fig. 11a presents an example of longer and shorter waves observed together. They are a set of bright and dim cloud bands that suggest wavelengths of ~150-200km and shorter waves with wavelengths of ~20-60km, indicated by the pairs of magenta arrows with solid heads and thin heads respectively. In this case the CIPS orbital strip achieves pretty satisfying capture of the waves because ~150-200km is still a short wavelength relative to the cross-orbit span (~900km) of the CIPS orbital strip. The albedo map reveals that the longer and shorter waves are nested together and the longer waves appear to have large amplitudes. The albedo power map in the lower panel shows that the wave power is primarily distributed in the upper-right and lower-left quadrants, which reflects the orientation of the wave ridges and troughs that are perpendicular to this.

The albedo power spectra for this particular case are shown in Fig. 11b and it reveals a -3.0 slope over the scale range of ~20-150km. The correspondence of $freq_{25}$=0.34 and albedo power=$25.0\times10^{-12}sr^{-2}$ makes this case being the closest to the medium albedo power category shown above, and the -3.0 spectral slope is also close to the -2.9. The spectra are not reliable in either the long wave limit (>150km) or the short wave limit (<20km). In the long wave limit the elliptical region does not capture enough repeats of ridge and trough and as a result the power spectra in this range often readily change when the CWT is applied to much expanded region (>400km) (not shown). In the short wave limit the measurement noise will contaminate the PMC signals. It is worth mentioning that the ~20-60km scale range focused in this study is for the shortest waves CIPS can resolve due to the ~5km spatial resolution and the signal noise levels (also see Randall et al., 2017). If $1.0\times10^{-6}sr^{-1}$ is taken as the threshold of the noise level in the CIPS measurements, the wave signatures will be systematically contaminated for the low albedo power category based on approximate amplitudes shown in the Fig. 6b above.

The albedo map and the power spectra in the presumed valid scale range (~20-150km) together indicate that longer and shorter waves are nested together with decreasing albedo power. This reminds us of the term "self-similarity" that is used to describe the possible fractal nature of the PMC albedo field with a fractal perimeter dimension of 1.3 (Brinkhoff et al., 2015; von Savigny et al., 2011). Self-similarity generally refers to the condition that the small and large scale structures resemble each other in morphology. We observed a hint of self-similarity in the gravity wave manifestation described by a -2.9±0.3 law of albedo power, but such an analogy is still preliminary and it requires a further investigation to confirm. It is also worth mentioning that the cascading power spectra from large to small scales (3000km-100km) of periodic PMC structures were also reported by Carbary et al. (2000) via analyzing the middle ultraviolet (210-252nm) images from the

Mid-course Space Experiment (MSX) (Carbary et al., 1994). However in their study no structures of <100km wavelengths were detected and the authors attributed this to the small amplitudes of the shorter wavelength waves that are not able to rise above the noise level.

6. **Conclusions**

A large ensemble of gravity waves resides in the PMCs and we aim to extract a universal law that governs the wave display throughout the full set of wave population. More specifically, we examined how wave morphology and the clarity level of display vary throughout the wave population manifested through the PMC albedo data. Higher clarity refers to more distinct exhibition of the features which often correspond to larger variances and better organized nature. Later we found that an analytic form of the albedo wavelet power spectra, i.e., $A \times (1/\text{wavelength})^{\alpha}$, precisely determines the level of display clarity, where the coefficient A and exponent $\alpha$ vary with different sub-groups of wave detections. This form of albedo power spectra is yielded because the gravity wave signatures in the PMCs are mostly quasi-periodic rather than strictly periodic and also because a hierarchy of scale ranges possess monotonically decreasing power density. The corresponding wavelet power spectra therefore do not exhibit any spectral peaks, especially when spatial average of the spectra has been conducted.

A gravity wave tracking algorithm is designed and applied to the PMC albedo data taken by the AIM CIPS instrument to obtain the gravity wave detections throughout the two northern summers in 2007 and 2010. The horizontal wavelengths in the range of ~20-60km are the focus of the study because they are the most commonly observed and readily captured in the CIPS orbital strips. An individual detection is carried out within an elliptical region of 400km along-orbit and 400×0.65km cross-orbit, and the center of the elliptical region is moved around within any given CIPS level-2 orbital strip by steps of halved axial lengths in both along- and cross-orbit directions to capture wave signatures at different locations and orientations. For a given location where the elliptical region is placed, a 1-D continuous wavelet transform (CWT) calculation is carried out along all 360° directions with 3° of intervals to obtain the wave power map enclosed in this region. The factor 0.65 in the cross-orbit direction is empirically chosen owing to the initial intent to better detect the concentric waves in the PMCs. This factor will not qualitatively affect the results of the wave tracking study.

The histograms of the albedo CWT power indicate that a majority of gravity waves of ~20-60km wavelengths reside in the lower albedo power and lower background brightness region, and the number density per bin of the albedo power shows an exponential decay toward high power values.

The cloud population is split between the dimmer and brighter groups using a defined threshold frequency of the bright cloud presence to examine the gravity wave manifestation in the dimmer and brighter backgrounds respectively. Within the elliptical region, if the bright cloud ($>25 \times 10^{-6}\text{sr}^{-1}$) frequency ($\text{freq}_{25}$) exceeds 2% then we will call it a brighter background. The threshold $25 \times 10^{-6}\text{sr}^{-1}$ is empirically chosen. Accordingly the dimmer backgrounds refer to the elliptical regions within which $\text{freq}_{25} < 0.02$. In this study we use both $\text{freq}_{25}$ and the mean albedo (within the elliptical regions) to describe the background PMC brightness.

The scatter plots of albedo power versus $freq_{25}$ and albedo power versus mean albedo (within the elliptical regions) both indicate that statistically albedo power monotonically increases with the PMC background brightness. In this paper we do not pursue what drives such an apparent relationship between the two variables. Rather, we resample the albedo power values to make them conform to a normal distribution based on an analytic form of the albedo power dependence on the background cloud brightness. A sequence of sectioning curves is used to regroup the wave detections. Via the resampling procedure we aim at extracting the law that controls the wave display beyond the apparent dependence of the albedo power on the background cloud brightness. In each resampling bin all brightness levels are included. It is worth noting that the monotonic relationship between the wave amplitude and the background cloud brightness does not seem to reflect the previously discovered driving mechanism between the gravity waves and the local PMC brightness, such as by Jensen and Thomas (1994) and Chandran et al., (2012). These authors suggested that long or short gravity waves eventually reduce the local cloud brightness level.

Sample cases are selected at the two tails and the peak of the normal distribution and at three brightness levels ($freq_{25}$=0, 0.4, and 0.8) to represent the full set of wave detections. The selections are made following the resampling sectioning curves on the scatter plot and are categorized as possessing the high, low, and medium albedo power. As is mentioned above, the resampling procedure is applied to examine the wave display beyond the apparent dependence of the albedo power on the background cloud brightness.

The albedo power spectra over scales ~20-60km for the representative cases follow a universal form of $A \times (1/wavelength)^{\alpha}$, where $\alpha$=-3.2, -2.9, and -2.5 and A=$1.49 \times 10^{-4}$, $1.45 \times 10^{-4}$, and $1.42 \times 10^{-4}$ for the high, medium, and low albedo power categories, respectively. The parameters A and $\alpha$ both take part in determining the overall power magnitude and the decay rate toward the smaller scales. It is worth noting that the three $\alpha$ values will undergo a minor systematic shift when different form of wavelets (e.g., Mexican hat) are adopted, but they will remain close to -3.0. Note that we have argued above that Morlet wavelet is an optimum choice in terms of reflecting both the periodicity and localization of the wave signatures. The overall larger power and the more rapid decay rate both lead to higher clarity of the wave display because the variance level will be higher and meanwhile the leading scale will be more dominant resulting in better organized nature. Each albedo power category possesses a consistent exponent (i.e., $\alpha$). As a result, via a simple adjustment on the coefficient A the power spectra between different brightness levels precisely collapse to each other and therefore a consistent level of display clarity will be achieved. The display clarity degrades substantially from high to low albedo power categories, which seems to suggest that there are wide spread and variable diffusive processes at the PMC height. The majority of the detected waves are straight waves or the interference of the straight waves regardless of the background brightness levels. Nevertheless, looking into more details we found that the wave signatures in the brighter background seem to exhibit mildly turbulent-like feature, suggesting that the wave patterns are less linear under this condition.

Exploration of longer wavelength gravity waves suggests that longer and shorter wavelength waves are likely nested together with decreasing albedo power. We may speculate that gravity waves of a hierarchy of scales from ~400-

500km to ~20-60km are likely equally widespread and possess exponentially decreasing albedo power density as the wavelength shortens.

Future work includes characterizing the coherency of the wave structures so that the algorithm can work effectively to identify specific wave events and wave morphology. After understanding the power law that governs the overall manifestation of the gravity waves in the PMCs, further identifying individual cases would be vital to eventually understand the mechanism of the gravity wave upward propagation from the source region to the site of display as the algorithm can also be applied to different altitude levels.

## Acknowledgements

This work was accomplished at the Center for Atmospheric Sciences, Hampton University, Hampton, Virginia. Funding for the AIM mission was provided by NASA's Small Explorers Program under contract NAS5-03132. The project is further supported by a NSF funding won in 2017 with award number 1651394. AIM CIPS data is available to public at http://lasp.colorado.edu/aim/download-data-L2.php. We thank the CIPS retrieval team for their tireless work on the CIPS data retrieval and for the well maintained and up-to-date status of the online download engine. We appreciate the data archiving team in Hampton University for keeping up the pace of data download onto the local server. We also greatly value the discussion and insights provided by the AIM science team members such as Drs. Mike Taylor and Yucheng Zhao during the course of this research work.

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

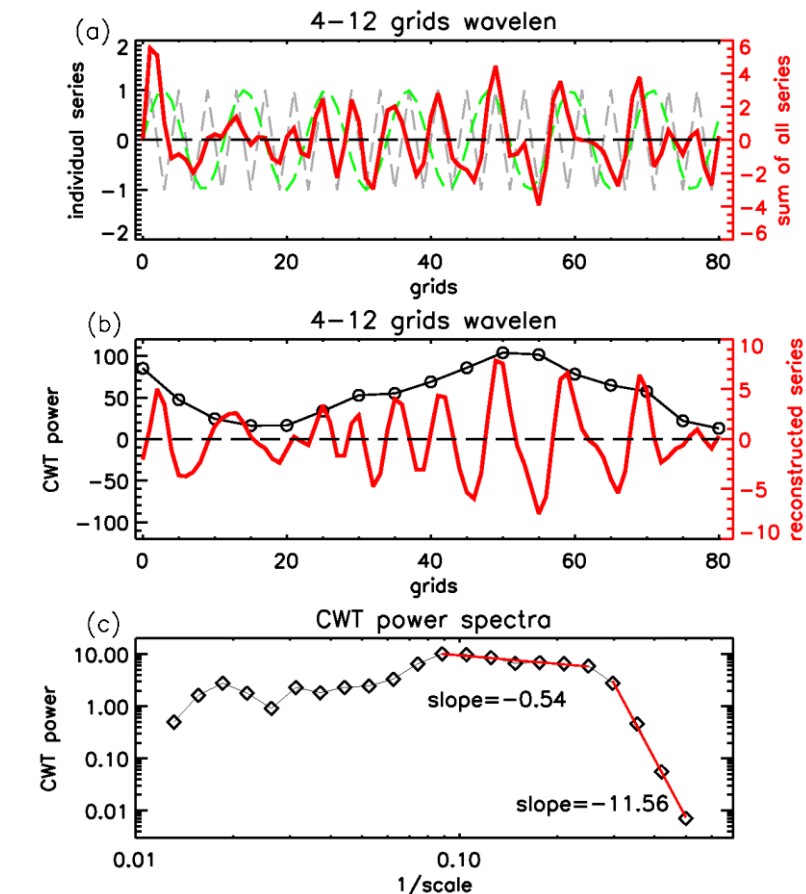

**Figure 1: Demonstration of the continuous wavelet transform (CWT) applied to an artificially created series. (a) The created series (red line) consisting of seven sinusoidal components with wavelengths (or scales) of 4.0, 4.7, 5.6, 6.7, 9.5, and 11.3 grid units, and each of them has the same amplitude of 1.0. The gray and green dashed lines are for the scales of 4.0 and 11.3 grids respectively serving as demonstration of the individual series. Note that the red curve has used the scale on the right axis which corresponds to a three times larger magnitude. (b) The total CWT power series (black with circles) and the reconstructed series (red line) using the wavelet coefficients and components. (c) The CWT power spectra. The slopes (i.e., red lines) are calculated over the seven scales used to create the series and the smaller scales that are present due to the non-orthogonal basis in the CWT calculation.**

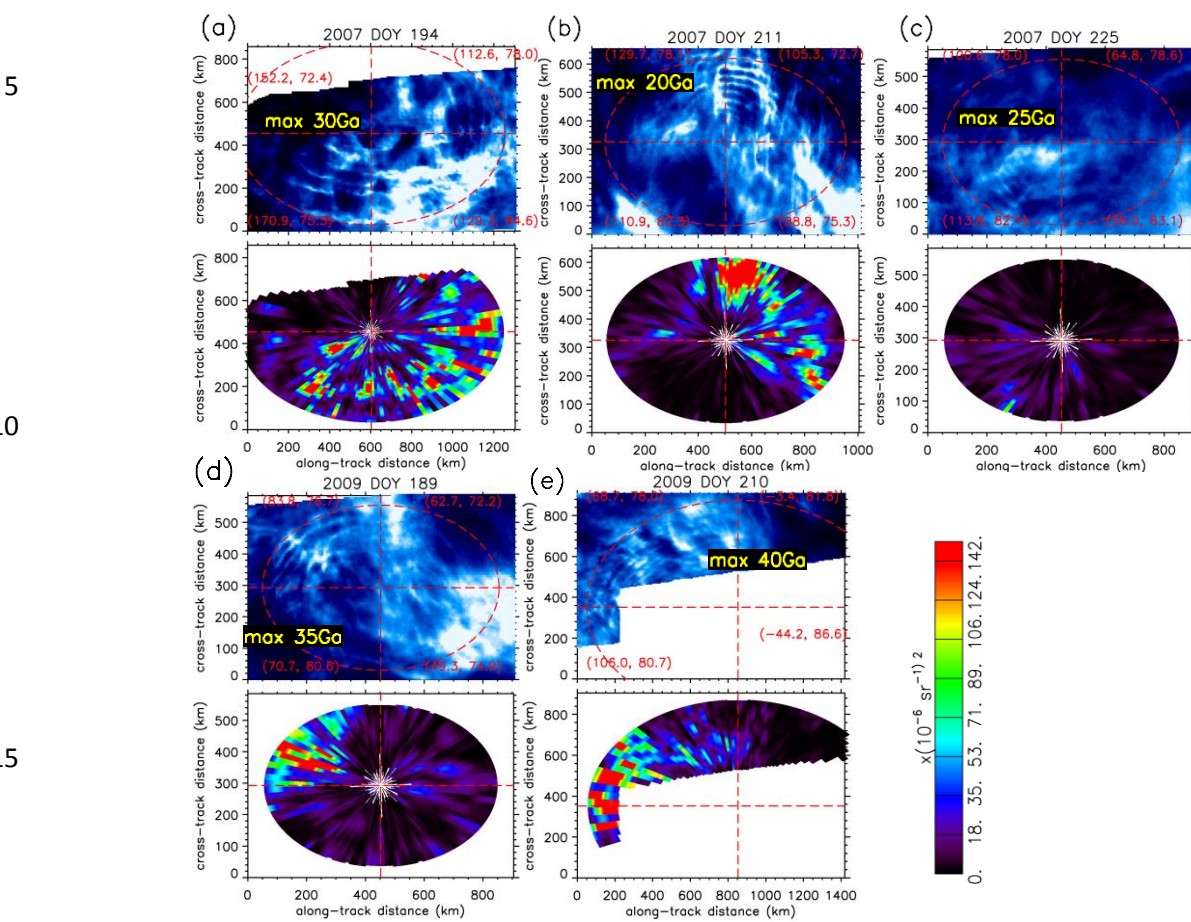

**Figure 2: Wave tracking algorithm applied to the concentric waves (a)-(e) in the CIPS orbital strips. Wave tracking is carried out within the elliptical regions. For each pair, the upper panel is the albedo and the lower panel is the albedo CWT wave power field by 3° angular bin × 5 km radial bin. Blue-white color scheme is used for the albedo maps with the white color representing the maximum albedo values indicated by the yellow legends, with 1 Ga=1.0×10⁻⁶sr⁻¹. The red numbers at the four corners are longitudes and latitudes. The rainbow color-bar for the CWT power is used universally in this paper.**

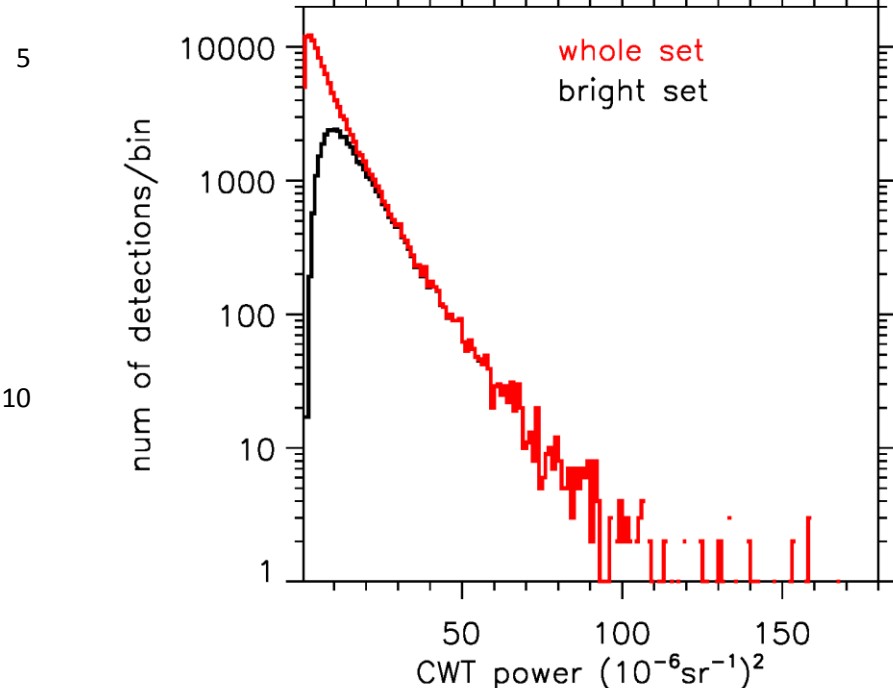

Figure 3: The histograms of the CWT power values obtained throughout the two northern summers in 2007 and 2010. Each CWT power value refers to an average within a given elliptical region. The black curve is for the brighter cloud group that corresponds to $freq_{25}$>2% within a given elliptical region and the red curve is for the full set of detections. The bin size within which we count the detection number is $1.0\times10^{-12}sr^{-2}$.

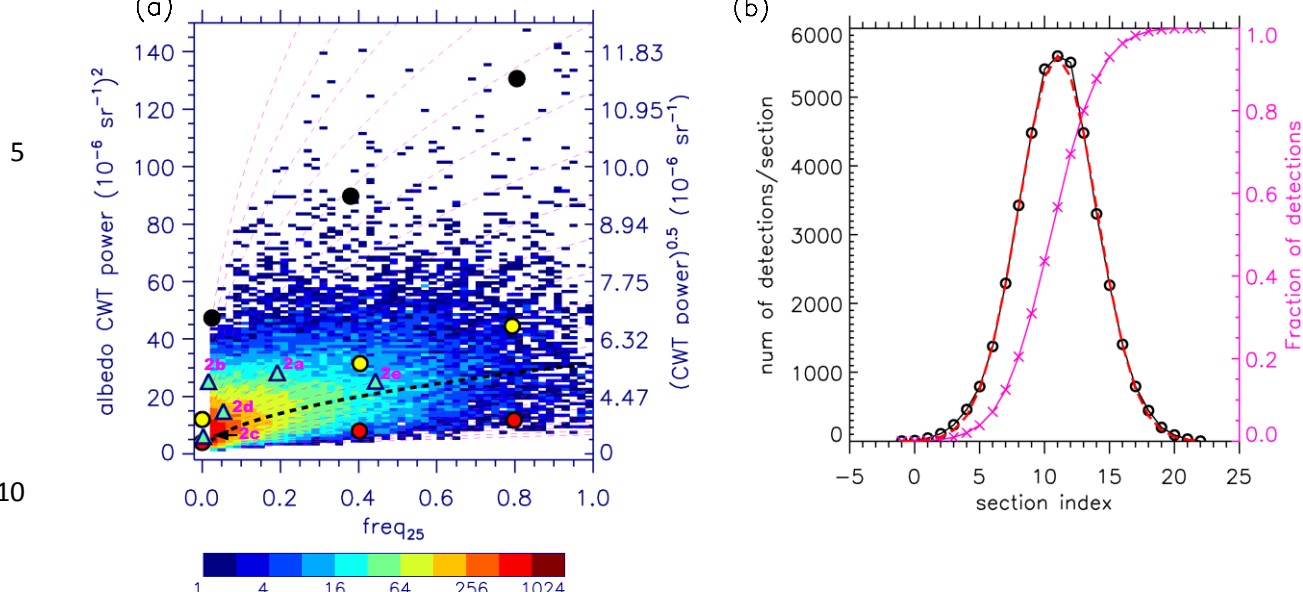

**Figure 4: (a) Scatter plot of the wave detections to reflect the relationship between the bright cloud frequency ($freq_{25}$) and the albedo CWT power. Only detections corresponding to the brighter clouds ($freq_{25}>0.02$) are included. The rainbow colors are the detection number density within each bin [$\Delta freq_{25}=0.02$ and $\Delta CWT= 1.0\times10^{-12} sr^{-2}$]. The dashed thick black lines roughly follow the peaks of the detection number density isopleths, serving to derive an analytic relationship between the cloud brightness and the CWT power. The turquoise triangles correspond to the concentric waves shown in Figure 2. The colored dots are selected roughly at the two tails and the peak of the normal distribution and at three brightness levels of $freq_{25}=0$, 0.4, and 0.8. These nine selections are representative to the full set of wave detections, to further demonstrate the wave exhibition in Figures 7-9. Using a sequence of analytic curves (magenta dashed) to resample the wave detections we obtain a normal distribution as is shown in (b). In (b) the black curve symbolled with the circles represent the obtained distribution from the left panel while the red dashed curve is the Gaussian fit. The magenta curve symbolled with crosses are the accumulated fraction of the data points.**

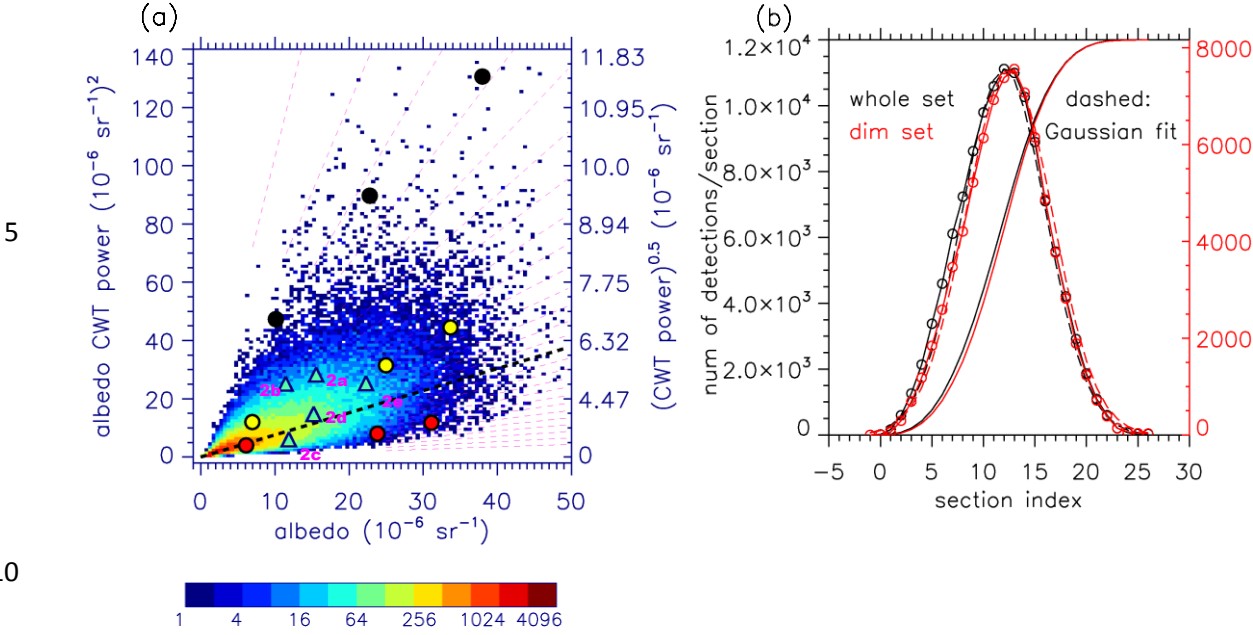

**Figure 5: (a) Same as Figure 4 except for using both dimmer and brighter sets of detections and use the mean albedo within the elliptical region as the horizontal axis. The bin of albedo is $0.5 \times 10^{-6}$ sr$^{-1}$. The turquoise triangles and colored dots are the same as in Figure 4. (b) The normal distributions for both the dimmer subset and the full set of wave detections. The curves without symbols are accumulated fraction of detections with a vertical axial range of 0-1.0 (not shown), which is the same as in Figure 4b.**

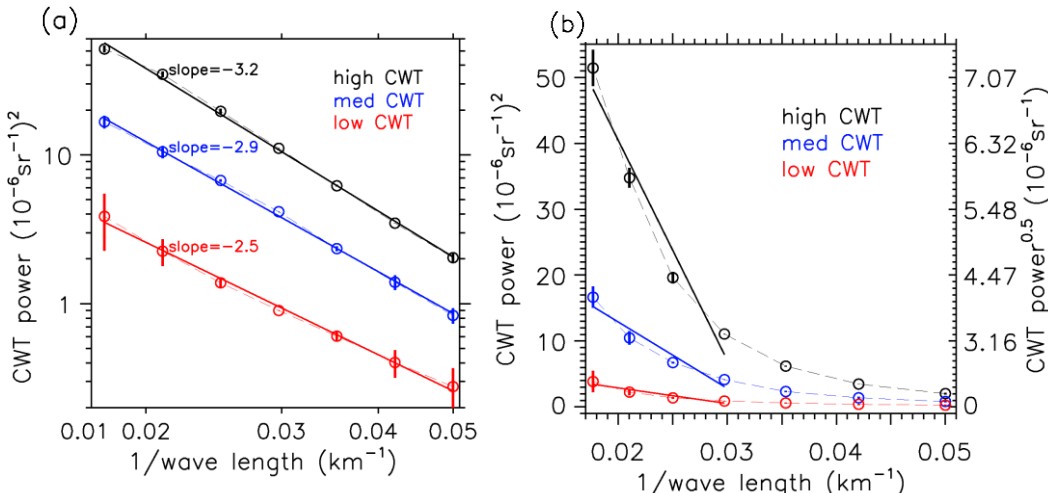

**Figure 6: Albedo CWT power spectra for the nine detections selected in Figure 4. The black, blue, and red colors correspond to the uppermost row (high albedo power), middle row (medium albedo power), and the lowermost row (low albedo power) of dots in Figure 4a respectively. For each albedo power category the CWT power spectra are normalized to collapse within the error bars as the 1-σ standard deviation. The exact match occurs at the medium scale (~33km wavelength or 0.03 spatial frequency) to obtain the mean slope. The thin dashed lines are original curves and the thicker solid lines are linear fitting lines. (a) Exponents of decay are -3.2, -2.9, and -2.5 for the three categories respectively. (b) Under linear axes it shows that higher albedo power and larger exponent correspond to more rapid decay of the albedo power toward smaller scales, reflected by the linear fitting lines for the first few dominant scales.**

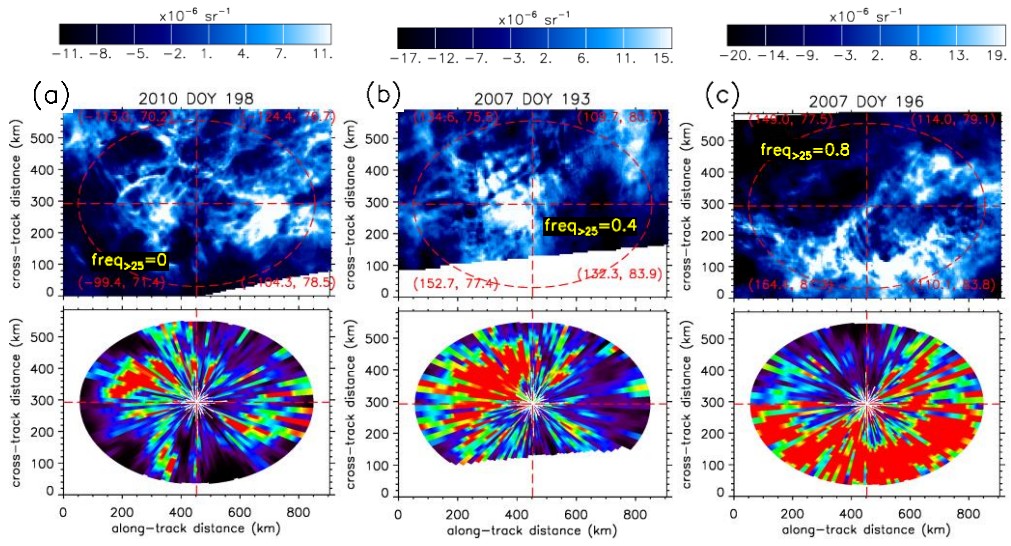

**Figure 7: Albedo maps and the albedo power maps corresponding to the high power category at three different brightness levels, each with the mean background albedo subtracted. The blue-white color scheme used linear RGB color code distribution. The albedo power maps used the same rainbow color bar as in Figure 2. Based on the factors used to make the albedo power spectra collapse between the different brightness levels, which are Ratio$_{0.4/0.8}$=1.44 and Ratio$_{0.0/0.8}$=3.13 respectively in this case (see Eqs. (1) and (2) in Sect. 5.1), the corresponding color bar maxima are reduced by factors √1.44=1.2 and √3.13=1.77, to achieve the same level of display clarity.**

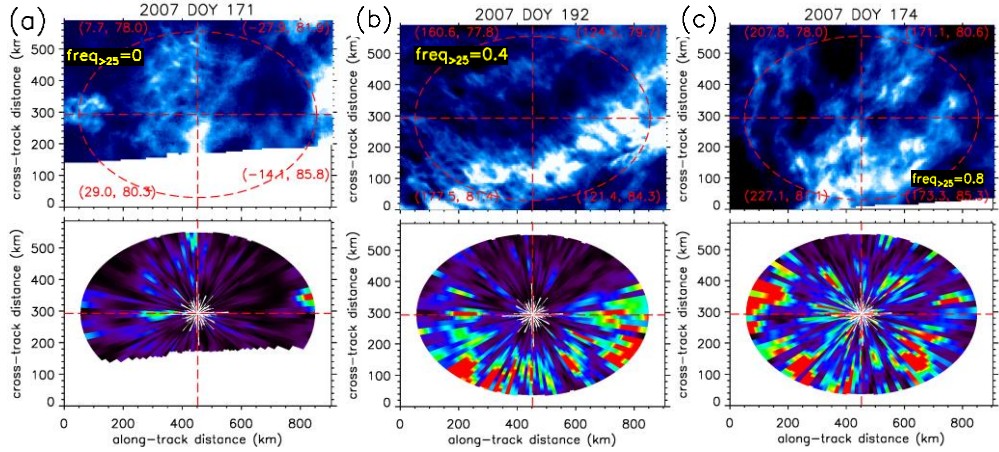

**Figure 8: Same as Figure 7 except for corresponding to the medium albedo power category. They are systematically more blurry than those in Figure 7. Same set of color-bars as in Figure 7 are used.**

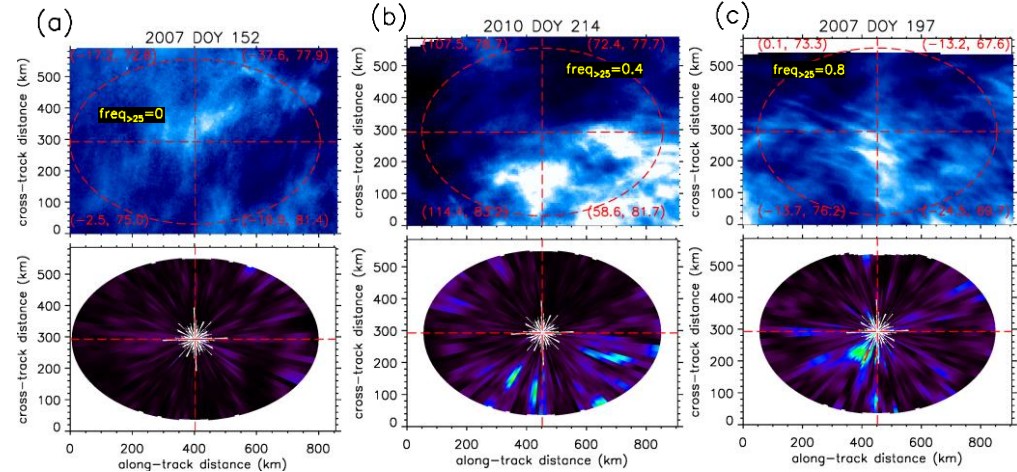

**Figure 9: Same as Figure 7 except for corresponding to the low albedo power category. They are further more blurry than what is shown in Figure 8. Same set of color-bars as in Figure 7 are used.**

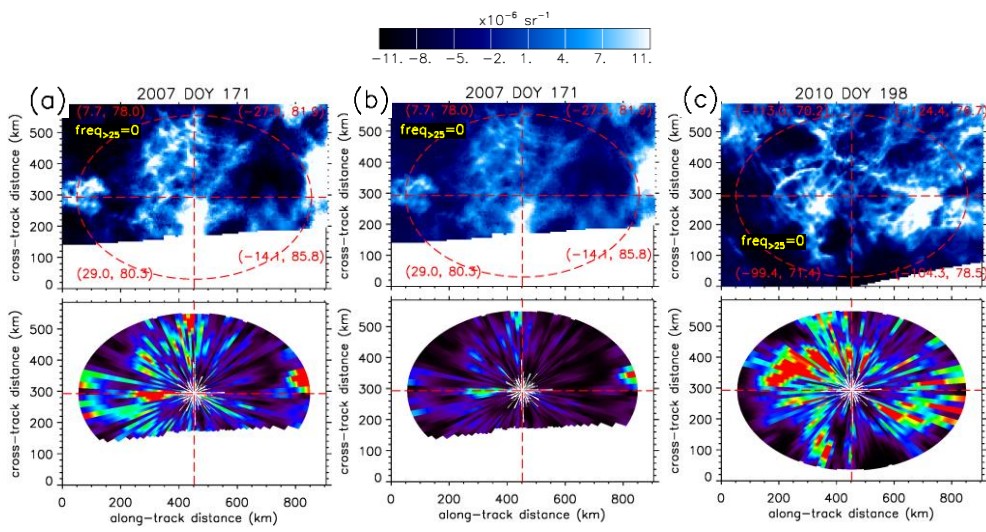

**Figure 10: (a) To demonstrate that amplifying the waves shown in the above Figure 8a (current Figure 10b) by a constant factor √2.778 (see text in Sect. 5.3) enhances its display clarity significantly. The enhanced display clarity is drawn closer toward what the above Figure 7a (current Figure 10c) shows. Note that Figure 7a belongs to the high albedo power category and has the highest level of display clarity.**

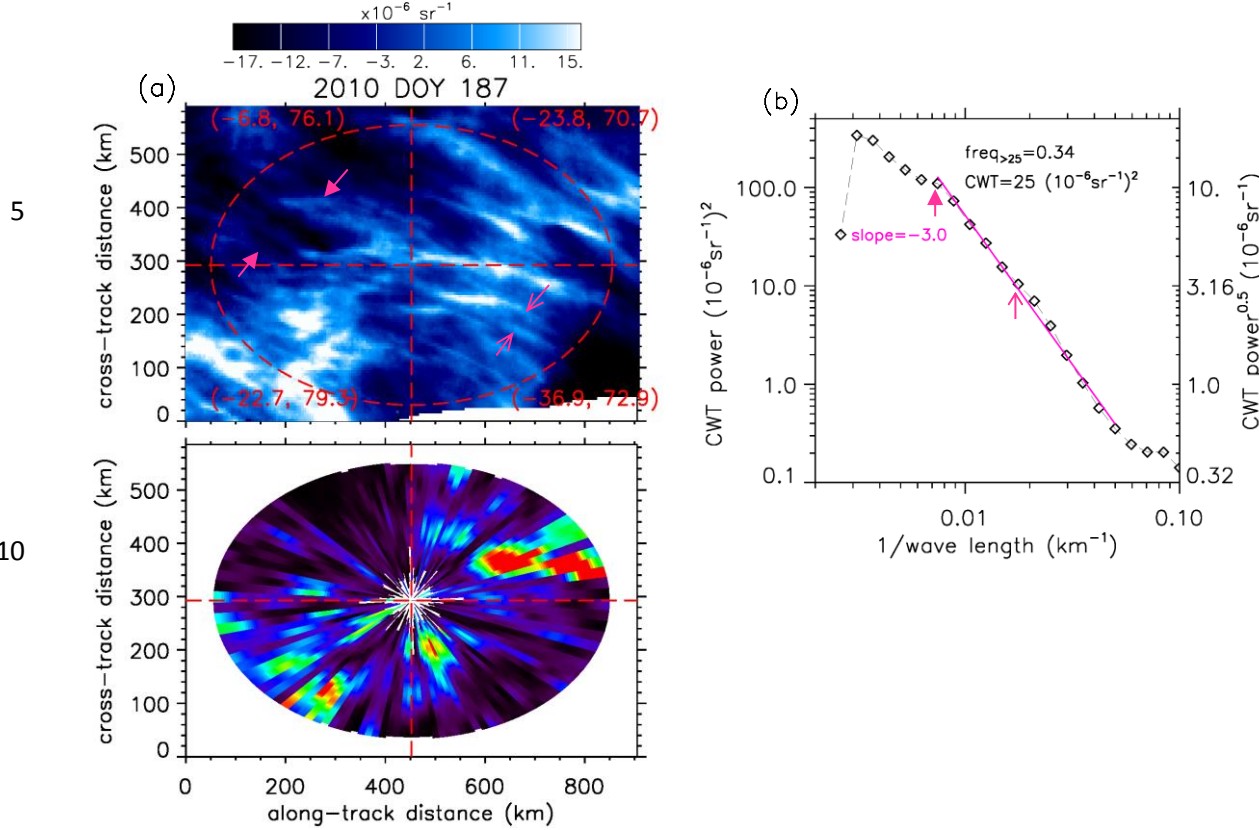

**Figure 11: Demonstration of longer and shorter waves nested together. (a) The albedo map and the corresponding albedo power map. The longer waves are roughly at ~150km wavelength (between solid arrows) and the shorter waves are roughly at ~40km wavelengths (between thin arrows). (b) The albedo power spectra with an exponent of -3.0.**