# Peer review of "Universal power law of the gravity wave manifestation in the AIM CIPS polar mesospheric cloud images"

_Atmospheric Chemistry and Physics, 2017_

## Referee Comment (RC1) · Anonymous Referee #1 · 31 Aug 2017

General comments:

This manuscript deals with the investigation of gravity wave signatures in polar mesospheric cloud (PMC) images observed with the CIPS instrument on the AIM spacecraft. More specifically, the authors attempt to determine a universal scaling law describing the scale dependence of the wave spectral power for the spatial scale range from about 20 to 60 km. The analysis approach is novel, as far as I can tell, and based on a directional 1-D wavelet analysis. The paper is overall well written, presents interesting results and should eventually be published in my opinion. I do ask the authors, however, to consider the following general and specific comments. My main concern is that

the analysis is based on several arbitrary assumptions and it is not clear to me how robust the obtained results (e.g., the derived spectral exponents) really are, and how they can/should be compared to similar studies based on different assumptions.

Here are some more general comments:

- The focus on wave signatures with wavelengths between about 20 and 60 km seems an unnecessary restriction in several parts of the manuscript, because many of the observed wave signatures have longer wavelengths (e.g. page 6, line 8 and the following lines).

- Wavelet power spectra are determined and analyzed in this study. In many studies on related subjects Fourier power spectra are analyzed. I'm wondering, whether the spectral exponents for wavelet and Fourier spectral are (necessarily) identical? The exponent will certainly depend on whether the wavelet power spectra are plotted as a function of wavelet scale or Fourier equivalent scale (see specific comment below). And perhaps the exponent depends somewhat on the mother wavelet used?

- "Wave tracking" and identification of gravity waves: The term "wave tracking" is used several times in the paper, but it seems no "tracking" of gravity waves is actually done. Spectral power in a certain scale range is used as a proxy for gravity wave activity, right? I'm wondering, whether enhanced spectral power is always an indicator for gravity waves? One can easily produce synthetic time series with similar scaling laws that have little to do with gravity waves. Perhaps some comments can be added along these lines of thought.

Specific comments:

Page 1, line 24: I suggest replacing "have played" by "play" in this sentence.

Page 2, line 21: "which is" -> "which are" ?

Page 3, section 2: Please mention what version of the CIPS data was used here. There may be different versions for the Level 0/1 and Level 2 data. All versions should

be mentioned.

Page 4, line 9: "The relevant scales for this study ARE"

Page 4, line 14: "These calculations will be carried out along all 360deg radial direction when being applied to CIPS PMCs"

Please mention explicitly for how many radial directions the analysis was carried out. Somewhere later you mention that this was done in 1 deg steps (page 10, line 2). I wonder how this is done specifically, and I think this should be described in more detail. You probably had to do a resampling (or projection) of the data onto each individual direction, which will also affect the spatial scales covered. I assume all of these effects were taken into account properly?

Page 4, line 19: "In a 6th-order Morlet wavelet (k=6) the scale s is almost precisely the period of the sinusoidal signal"

This is correct, but please mention whether you used the wavelet scale or the Fourier equivalent scale for the following analysis and discussions. This choice may/will affect the spectral exponents derived later.

Page 5, line 5: "Over the seven relevant .."

I suggest keeping the sign of the slope, i.e., writing " -0.54".

Page 5, line 6: "later" -> "latter" ?

Same sentence: Looking at panel a) of Fig. 1, one observes that the dominant (apparent) period changes from about 5 grid units between 20 and 40 to about 10 grid units at 50 to 60, i.e. one would not expect to observe a clear spectral peak in the averaged power spectrum. In the original power spectrum (not averaged over the grid points), one would be able to identify this shift in the dominant period, i.e., you may also show the full power spectrum (but this is not really necessary in my opinion).

Page 5, line 13: ".. along all radial directions"

As above, it would be good to mention for how many directions this was done.

Page 5, line 15: "In addition, performing the CWT along all radial directions will efficiently capture the waves of all orientations without having to perform systematic CWT calculations along-orbit and cross-orbit."

I don't really understand the logic behind this statement (and I'm probably missing something here). Doing the analysis along all radial directions requires many more calculations than doing it only along-orbit and across-orbit, right?

Page 5, lines 17/18: It is not fully clear to me why it is necessary to use a 2-D FFT routine to carry out a 1-D wavelet transform?

Page 6, line 19: I suggest deleting "that" in "some physical process THAT yet to be unraveled"

Page 7, line 4: ".. carried out throughout the two northern summers in 2007 and 2010"

Was there a specific reason to limit the analysis to these two summers? If yes, it would be good to mention it.

Page 7, line 6: "We split the cloud population into two subsets .."

I'm sorry, I read this and the following sentences several times, but I don't fully understand the exact difference between the two groups. Can you please make this distinction more transparent. Also, it is not entirely clear to me, what freq_25 means. It seems, these sentences mix different definitions in a non-trivial way which makes it difficult to understand them.

Page 8, line 7: "We next adopt an analytic form to parameterize the relationship between the albedo power and freq_25 in order to resample the wave detections into a consistent normal distribution."

This is certainly something one can do, but this step (and the sectioning performed) seems quite arbitrary to me. Why is a normal distribution really needed? My concern

is that the analysis is based on several arbitrary choices, which raises the question how robust the obtained results (e.g., spectral slopes) are and how they can be compared to other studies that are based on different assumptions.

Page 9, lines 3 – 7: As far as I can tell it is not mentioned how the line sectioning is done. Is it based on an analytic expression?

Page 10, line 28 and line 29: "noises" -> "noise" ?

Page 11, line 17: "mildly turbulent-like"

Can you express this in a more objective way. Looking at the Figures I'm not sure, what exactly is meant here. I also wonder, whether a conclusion like that can be drawn based on 3 cases only?

Page 12, line 7: "three-colored" -> "three colored"

Page 12, line 12: "it plays a minor importance." -> "it is of minor importance." or "it plays a minor role." ?

Page 12, line 29: "wave lengths" -> "wavelengths"

Page 13, line 21: Suggest writing "fractal perimeter dimension" rather than just "fractal dimension", because there are different fractal dimensions.

Page 15, line 13: "wave length" -> "wavelength"

Page 15, line 29: spaces before "Randall" and "von Savigny" missing

Page 16, line 6: delete extra space after last author

Page 16, line 8: add space before paper title.

Figure 1, caption, line 2: add space in "and11.3"

Figure 2, caption, line 5: "x5 km" -> "x 5 km" ?

Figure 3, ordinate label: "num of detections / bin"

It's not clear (at least to me) what "bin" refers to here.

Figure 4, caption, line 3: "The dashed thick black lines roughly follow"

There only seems to be a single thick black line, not multiple lines.

Figure 6: It would be good to determine and present uncertainty estimates for the spectral slopes.

[Figure]

---

## Referee Comment (RC2) · Anonymous Referee #2 · 31 Aug 2017

In their paper the authors investigate the modulation of polar mesospheric clouds (PMCs) by gravity waves of horizontal wavelengths in the range 20 to 60km. A novel gravity wave tracking algorithm is applied to PMC observations by the AIM Cloud Imaging and Particle Size (CIPS) instrument. The wave detections are resampled depending on the background cloud brightness and the wave power obtained from the detection algorithm. By doing so, the wave detections follow a normal distribution, and it is possible to derive general properties of the whole distribution of wave detections. It is found that the wave power decays toward small scales with an exponent between -2.5 (for low spectral power events) and -3.2 (for strong events). Further, weak events are more blurry and seem to be affected by turbulence.

[Figure]

Overall, this study is an important step forward in the characterization of the interaction of gravity waves with PMCs. Particularly, the resampling of wave detections allows to deduce general properties of the wave distribution. The paper is well written, and the topic is of broad interest for the readership of ACP. The paper is therefore recommended for publication in ACP. There are only a few minor comments that should be addressed before publication.

MINOR COMMENTS:

(1) The introduction is too much focused on the very short horizontal wavelength gravity waves and their effect on PMCs. Only in Sect.5 and in the conclusions it is mentioned that also gravity waves with quite long horizontal wavelengths (of several hundred km and more) modulate PMCs, and that their wave power can be even stronger than that of the short scales addressed in the current paper. This information, however, should already be given in the introduction.

Therefore after p2, line 11 you should add the following information: It is also known that long horizontal scale gravity waves with wavelength of hundreds of km and longer exist in the mesosphere / lower thermosphere region (for example, Ern et al., 2011; Wachter et al. 2015). Corresponding variations are found in PMC brightness (for example, Carbary et al., 2000), and it was argued by Chandran et al. (2012) that observed PMC patterns may be caused by a superposition of large scale (>300km) and small scale (<300km) gravity waves with the large scale long period gravity waves producing the most significant increases in albedo.

References:

Carbary, J. F., Morrison, D., and Romick, G. J.: Transpolar structure of polar mesospheric clouds, J. Geophys. Res., 115, 24763-24769, 2000.

Chandran, A., D. W. Rusch, G. E. Thomas, S. E. Palo, G. Baumgarten, E. J. Jensen, and A. W. Merkel (2012), Atmospheric gravity wave effects on polar mesospheric

clouds: A comparison of numerical simulations from CARMA 2D with AIM observations, J. Geophys. Res., 117, D20104, doi:10.1029/2012JD017794.

Ern, M., P. Preusse, J. C. Gille, C. L. Hepplewhite, M. G. Mlynczak, J. M. Russell III, and M. Riese (2011), Implications for atmospheric dynamics derived from global observations of gravity wave momentum flux in stratosphere and mesosphere, J. Geophys. Res., 116, D19107, doi:10.1029/2011JD015821.

Wachter, P., Schmidt, C., Wuest, S., and Bittner, M.: Spatial gravity wave characteristics obtained from multiple OH(3–1) airglow temperature time series, J. Atmos. Solar-Terr. Phys., 135, 192-201, 2015.

(2) p2 l28+29 and p3 l10 Here you are talking of "larger scale features". Please be more specific. Could this be larger scale gravity waves, or are there other effects?

(3) p3 l13+14: Here you mention that in the current paper no horizontal filtering is applied like in Chandran et al. (2010) where polynomial smoothing is used as a background, and only remaining fluctuations are analyzed. Still, your analysis technique focuses on a (narrow) range of horizontal wavelengths. This means that indirectly high pass filtering is applied by limiting the wavelet to the desired range of 20-60km (see p4 l10). The main difference to Chandran et al. (2010) is that you are using a sharp cutoff, and the spectral characteristics of the remaining wave spectrum will not be altered by the filter characteristics. I think this is an important point that should be mentioned more clearly.

(4) p5 l13 Please be more specific. What is the angular step/resolution used? Later in the conclusions, I learned that the step-width is 3deg. However, this information should be given also on p5 l13.

(5) Fig.4a; Sect.4.3, p7 bottom Since the resampling of wave events is an important step in your analysis, this should be briefly illustrated by an example. Suggestion:

If in Fig.3 an event would occur in the histogram (black curve) with 50% of the values

at weaker CWT power, this event would obtain a value of freq25=0.5, and we find this event in Fig.4a at the coordinates (0.5, albedo power of the event).

(6) p10, l16 and p23, caption of Fig.6: The normalization procedure was still not fully clear to me. Could you elaborate a bit more on this? Is one of the three brightness levels taken as a reference, and the others are normalized to match the reference? Is this normalization derived for a given wavelength position? How are the error bars in Fig.6 obtained?

TECHNICAL COMMENTS:

(1) p1 l10 Gravity wave display morphology and clarity level -> Gravity wave display, morphology, and clarity level

(2) p21, Fig.4 panels (a) and (b) overlap, please allow for more space between them

(3) p21, caption of Fig.4, about Fig.4b Please mention that the red dashed curve represents a Gaussian fit (normal distribution) to the data (black dots), while the thin black curve is a polygon that connects the data points. The magenta crosses represent normalized integrals over the number of detections below the respective analytic curves in Fig.4a.

Still, the following was still unclear to me: What is the magenta line in Fig.4b? Is this a fit through the magenta crosses?

---

## Referee Comment (RC3) · Anonymous Referee #3 · 12 Sep 2017

General Comments:

This paper presents an analysis of data from the Cloud Imaging and Particle Size (CIPS) instrument on the Aeronomy of Ice in the Mesosphere (AIM) satellite that focuses on the potential to extract gravity wave information from the CIPS measurements. CIPS measures backscattered radiance at 265 nm, which provides information about the background atmosphere (excluding PMCs) at approximately 50 km altitude. An algorithm is presented to identify gravity waves with horizontal scales between ~20-60 km. Power spectra of the identified waves are determined, as well as the importance of the background albedo brightness.

The approach to identifying and characterizing gravity waves presented in this paper is successful in some situations. However, the ability to distinguish waves (and thus study their morphology) is clearly dependent on the background albedo intensity. The presence of multiple waves in any scene, with different scale lengths and orientations, is also a challenge for classification. One suggestion would be to investigate the use of image processing algorithms to sharpen the contrast of any wave-like features before applying the detection algorithm.

It is puzzling that the recent paper on CIPS gravity wave observations by Randall et al. [2017] (Geophys. Res. Lett. 44, 7044-7052) is not addressed in this manuscript. Four of the co-authors on this manuscript are also authors on the Randall et al. paper. Since the Rayleigh Albedo Anomaly (RAA) data used in the Randall et al. paper are also available at the AIM web site, I think it is essential that this manuscript discusses how their approach compares with the RAA method.

Some additional specific comments are listed below.

Specific Comments:

1. p. 4, lines 21-22: It would be helpful to note that the Y-axis scale for the red curve in Figure 1(a) is a factor of 3 larger than the scale for the individual series.

2. p. 4, line 31: If the individual component time series each have a maximum amplitude of $\pm 1$ unit, how can the reconstructed time series shown in Figure 1(b) have an amplitude greater than -7 at grid = 55?

3. p. 5, line 13: What is the angular step used to select "all radial directions"?

4. p. 5, lines 31-33: You can't call concentric waves "characteristic" in the first sentence and "extremely rare" in the next sentence. These terms have very different meanings.

5. p. 6, lines 23-25: Are the straight wave patterns present in the lower right quadrant of Figure 2(d)? It would be good to see more examples of straight waves, which are presumably much more common.

6. p. 7, lines 17-19: This exponential behavior is also seen in g-distribution plots of PMC brightness and ice water content (e.g. DeLand and Thomas [2015]).

7. p. 7, lines 21-23: Interannual variability of the g-distribution slope of SBUV PMC ice water content is discussed by DeLand and Thomas [2015].

8. p. 7, lines 29-31: What is the term "laminated" intended to mean here?

9. p. 9, line 1: Why is a different analytic form needed?

10. p. 10, lines 8-10: The fits for the different power categories seem to have almost identical slopes in the left-hand panel (log scale), despite the changes in quoted slope value. Is everything correct here?

11. p. 11, lines 11-13: This is another instance where straight waves seem to be more common, which raises the question as to why there is so much initial emphasis placed on concentric waves.

---

## Author Comment (AC1) · 16 Sep 2017

This reply and reply-figs, and the track-changed paper are all in this PDF file.

^^^^^^^^^^^^^^^^^^^^^^^^^^^^^^^^^^^^^^^^^^^^^

I do ask the authors, however, to consider the following general and specific comments. My main concern is that the analysis is based on several arbitrary assumptions and it is not clear to me how robust the obtained results (e.g., the derived spectral exponents) really are, and how they can/should be compared to similar studies based on different assumptions.

Specific questions will be responded at the corresponding locations but in this answer some of the seemingly arbitrary aspects are explained as follows:

1. An elliptical region with a specific axial ratio of 0.65 and along-axial radius of 400km.

400km radius is chosen empirically because we try to avoid including different nature of the waves within one elliptical region and yet the region should be large enough to include the main fraction of a given wave packet. We do however understand that gravity waves in the PMCs are highly complex and there is not a strict criterion to separate different events.

2. The scales are limited to 20-60km.

Please see the next reply for the choice of 20-60 scale range.

3. Robustness of the results:

(a) The roughly -3.0 spectral slope or exponential is robust.

(b) The slightly weakening slope with the decreasing wave power is also robust. However, using a different mother wavelet the slopes show minor differences.

Nevertheless, we consider Morlet wavelet be the most reasonable mother wavelet compared to Mexican hat. We will later show the results of Mexican hat and explain why.

In summary, for a complex entity of waves we made a choice to attack the problem to extract a common law as much as we can, which is the rationale of this study. But the reviewer's concern is fully legitimate because if the results vary strongly with the choice of the wavelet or any other given condition the goal of this research will not be served.

Here are some more general comments:

- The focus on wave signatures with wavelengths between about 20 and 60 km seems an unnecessary restriction in several parts of the manuscript, because many of the observed wave signatures have longer wavelengths (e.g. page 6, line 8 and the following lines).

There are several reasons that 20-60km is chosen:

1. CIPS does not effectively detect waves with wavelength<20km. See the track-changed page 14.

2. The short wavelength gravity waves are the most readily observed in the CIPS PMCs. As for the threshold at 60km, it happens to fall onto a specific scale in the CWT spectrum, which is why it is 60km rather than 50km or 70km. See the last paragraph of the track-changed page 4.

3. Larger scales (i.e., >60km) have higher power and it will substantially change the spatial power distribution. In this case we will not see the good correspondence between the wave features in the albedo maps and the wave power maps as is shown in our presented results.

> We have done an experiment including larger components in the spatial power maps and found that the power spatial distribution features are flat and broad because the region is too small to comfortably including the larger scales (not shown in this reply).

4. The Fig.11 of this paper included the larger scales and also confirmed a roughly -3.0 slope.

- Wavelet power spectra are determined and analyzed in this study. In many studies on related subjects Fourier power spectra are analyzed. I'm wondering, whether the spectral exponents for wavelet and Fourier spectral are (necessarily) identical? The exponent will certainly depend on whether the wavelet power spectra are plotted as a function of wavelet scale or Fourier equivalent scale (see specific comment below). And perhaps the exponent depends somewhat on the mother wavelet used?

This is a very important concern that must be addressed. Wavelet transform rather than FFT is used because the FFT basis vectors are not the best ruler for the localized yet quasi-period features. Please look at Reply-Fig-1. Morlet is the best wavelet because it resembles FFT locally. For example, Mexican hat wavelet has broader FFT spectrum and show fuzzier features spatially but smoother features spectrally compared to Morlet. As we have seen from Reply-fig-1, FFT possesses a highly fluctuating power spectrum because its basis vectors are over the full space and therefore do not reflect the local variability well.

However, given the differences between the different basis vectors, FFT, Morlet, and Mexican hat convey the main message consistently (Reply-Fig-1 bottom panel).

The redone results using the Mexican hat wavelet are shown in Reply-Fig-2. The albedo power spectra for Morlet and Mexican hat both roughly follow the same -3.0 law, and meanwhile the slope also weakens with decreasing wave power levels for the Mexican hat results.

There are minor differences between the two results but from Reply-Fig-1 we learned that Mexican hat wavelet is not as good a choice as the Morlet wavelet in terms of naturally reflecting the gravity wave features because the Mexican hat does not reflect the periodicity well.

- "Wave tracking" and identification of gravity waves: The term "wave tracking" is used several times in the paper, but it seems no "tracking" of gravity waves is actually done. Spectral power in a certain scale range is used as a proxy for gravity wave activity, right? I'm wondering, whether enhanced spectral power is always an indicator for gravity waves? One can easily produce synthetic time series with similar scaling laws that have little to do with gravity waves. Perhaps some comments can be added along these lines of thought.

The term "wave tracking" is a general term, meaning tracking all existing gravity wave displays in the CIPS PMCs. Please see the track-changed page 4 lower paragraph for a text revision.

In addition, in my opinion, tracking some of the waves, even not all scales are included, can be called wave tracking. The 20-60km wavelength wave structures are the most readily observed wave features in CIPS. I personally consider wave tracking is a fairly broad term that has less specific meaning than the term "wave detection".

The reviewer has a good point that high variance or power does not necessarily mean that there is a wave structure. True, we can purposely form a synthetic series that bears high lower. But we took advantage of the fact that in the Earth atmosphere summer mesopause region there is a high population of quasi-periodic gravity wave structures. It could be a turbulent structure although it is relatively rare. In our case it cannot be anything else. For example, it cannot be anything like vortex filaments that regularly appear in the polar winter stratosphere.

Strictly speaking, the coherency, or phase relation, is the indicator of the wave presence. We put this into the future work plan. Some text is inserted at the end of conclusion on the track-changed page 16 to address this intent.

Given the theoretical importance of the phase relation, we found that in CIPS most high albedo power is characterized by the quasi-periodic gravity wave features. For example, in Reply-Fig-4 it shows that the albedo map that corresponds to the largest albedo power (over 2007 and 2010) exhibits a set of neat straight wave pattern.

Specific comments:

Page 1, line 24: I suggest replacing "have played" by "play" in this sentence.

It is revised as suggested on the track-changed page 1.

Page 2, line 21: "which is" -> "which are"?

It is reworded on track-changed page 2.

Page 3, section 2: Please mention what version of the CIPS data was used here.

CIPS version "4.20" is now included on track-changed pages 3-4.

There may be different versions for the Level 0/1 and Level 2 data. All versions should be mentioned.

A small paragraph is inserted at the end of data section on track-changed page 4 with Scott Bailey et al.'s 2009 paper cited as another reference.

Page 4, line 9: "The relevant scales for this study ARE"

It is reworded on track-changed page 4.

Page 4, line 14: "These calculations will be carried out along all 360deg radial direction when being applied to CIPS PMCs" Please mention explicitly for how many radial directions the analysis was carried out. Somewhere later you mention that this was done in 1 deg steps (page 10, line 2). I wonder how this is done specifically, and I think this should be described in more detail.

It is a 3° step. A few sentences describing the rationale of this choice are inserted on the track-changed page 6 to better describe the resampling process.

You probably had to do a resampling (or projection) of the data onto each individual direction, which will also affect the spatial scales covered. I assume all of these effects were taken into account properly?

Please look at the same revision mentioned right above. Yes, resampling is done to project the original CIPS data onto the radial and angular directions, and it is is properly done to not lose details yet produce sufficiently smooth images.

Page 4, line 19: "In a 6th-order Morlet wavelet (k=6) the scale s is almost precisely the period of the sinusoidal signal" This is correct, but please mention whether you used the wavelet scale or the Fourier equivalent scale for the following analysis and discussions. This choice may/will affect the spectral exponents derived later.

This has been addressed in the general comments part. Please check on Reply-Fig-1. I copy the answer in the previous part as follows:

"Morlet is the best wavelet because it resembles FFT locally. For example, Mexican hat wavelet has broader FFT spectrum and show fuzzier features spatially but smoother features spectrally compared to

Morlet. As we have seen from Reply-fig-1, FFT possesses a highly fluctuating power spectrum because its basis vectors are over the full space and therefore do not reflect the local variability well.

However, given the differences between the different basis vectors, FFT, Morlet, and Mexican hat convey the main message consistently (Reply-Fig-1 bottom panel).

The redone results using the Mexican hat wavelet are shown in Reply-Fig-2. The albedo power spectra for Morlet and Mexican hat both roughly follow the same -3.0 law, and meanwhile the slope also weakens with decreasing wave power levels for the Mexican hat results.

There are minor differences between the two results. From what we've learned from Reply-Fig-1, Mexican hat wavelet is not as good a choice as the Morlet wavelet in terms of naturally reflecting the gravity wave features because the Mexican hat does not reflect the periodic nature well."

Page 5, line 5: "Over the seven relevant .." I suggest keeping the sign of the slope, i.e., writing " -0.54".

Minus sign is included and the sentence is reworded. See track-changed page 5.

Page 5, line 6: "later" -> "latter" ?

It is corrected on track-changed page 5.

Same sentence: Looking at panel a) of Fig. 1, one observes that the dominant (apparent) period changes from about 5 grid units between 20 and 40 to about 10 grid units at 50 to 60, i.e. one would not expect to observe a clear spectral peak in the averaged power spectrum. In the original power spectrum (not averaged over the grid points), one would be able to identify this shift in the dominant period, i.e., you may also show the full power spectrum (but this is not really necessary in my opinion).

True. The spectra are averaged over space because the wavelet spectra also are function of geometric space. In this research we do not go into the shorter spatial ranges to search the local wavelet spectral peaks. Such peaks always exist as is shown in the spectra map of a random series (see Reply-Fig-1 third row).

Fig. 1 has another fold of meaning which is, several equal-amplitude FFT components combined produce a quasi-periodic signal but among these scales not a single scale stands out.

But, 20-60 km is after all a narrow wavelength range and if put into a very wide scale range such as 10-300km there will be a broad peak that covers 20-60km.

Page 5, line 13: ".. along all radial directions" As above, it would be good to mention for how many directions this was done.

The resampling is now detailed on the track-changed page 6.

Page 5, line 15: "In addition, performing the CWT along all radial directions will efficiently capture the waves of all orientations without having to perform systematic CWT calculations along-orbit and cross-orbit."
I don't really understand the logic behind this statement (and I'm probably missing something here). Doing the analysis along all radial directions requires many more calculations than doing it only along-orbit and across-orbit, right?

Doing calculations along-orbit and cross-orbit requires two sets of calculations (therefore two fields) but along all radial directions by our design only gives us one set of calculation (therefore one field).

We wish to see a one-to-one correspondence between the albedo map and the albedo power map. Our calculation is straightforward and contains the main information of the wave power. A full 2D FFT

spectrum would give us many sets of spectra. Please see Reply-Fig-3 for several selected sets of 2D FFT basis vectors.

Page 5, lines 17/18: It is not fully clear to me why it is necessary to use a 2-D FFT routine to carry out a 1-D wavelet transform?

What it says is that doing our 1-D CWT along all radial directions is a short version of what the 2D FFT would convey. FFT routine is not used in this research.

Page 6, line 19: I suggest deleting "that" in "some physical process THAT yet to be unraveled"

It is corrected on the track-changed page 7.

Page 7, line 4: ".. carried out throughout the two northern summers in 2007 and 2010"

Was there a specific reason to limit the analysis to these two summers? If yes, it would be good to mention it.

A short answer, no (specific reason). But we do have a few considerations to do so. We wish to choose two years rather than just one in case there is any strong inter-annual variability. But overall, the current stage of the study is not operational therefore not all years are performed. It is time consuming to do them all. If any reason at all, 2007 has been used by many previous researchers and we add 2010 as another year that is some time apart from 2007 just in case retrievals in the early years have shared some common issues.

Page 7, line 6: "We split the cloud population into two subsets .."

I'm sorry, I read this and the following sentences several times, but I don't fully understand the exact difference between the two groups. Can you please make this distinction more transparent. Also, it is not entirely clear to me, what freq_25 means.
It seems, these sentences mix different definitions in a non-trivial way which makes it difficult to understand them.

The two groups are very different. One is the dimmer group and the other is the brighter group. We are particularly interested in the wave manifestation in the brighter group because most previous case studies are from the dimmer cloud environment. $freq_{25}$ is a parameter empirically defined to separate the dimmer and brighter cloud groups. $freq_{25}$ is defined as the fraction of cloud measurements with albedo>$25 \times 10^{-6} sr^{-1}$. The larger $freq_{25}$ is, the stronger the overall brightness level is within a given elliptical region.

Bright cloud frequency is a better index than the mean albedo within the elliptical region in terms of separating the bright and dim clouds.

Page 8, line 7: "We next adopt an analytic form to parameterize the relationship between the albedo power and freq_25 in order to resample the wave detections into a consistent normal distribution."
This is certainly something one can do, but this step (and the sectioning performed) seems quite arbitrary to me. Why is a normal distribution really needed?

$freq_{25}$ has no effect on the power slopes and it wasn't used in the power spectrum calculation.

Normal or Gaussian distribution indicates that it is free of the apparent impact of the brightness so that we are able to study the cloud gravity power spectra as an independent statistical variable.

My concern is that the analysis is based on several arbitrary choices, which raises the question how robust the obtained results (e.g., spectral slopes) are and how they can be compared to other studies that are based on different assumptions.

So far all the seemingly arbitrary aspects have been addressed as I believe. Also the power exponents (slopes) have been calculated using a different mother wavelet as is shown in Reply-Fig-2 (see above).

As for other studies, as far as we know there aren't similar attacking points yet to investigate the PMC gravity waves. Most previous studies focused on detecting wave peaks for particular cases or study the wave propagation or driving mechanisms.

Page 9, lines 3 – 7: As far as I can tell it is not mentioned how the line sectioning is done. Is it based on an analytic expression?

It was described in the paragraph started with "A set of square root sectioning curves" on the track-changed page 9.

Page 10, line 28 and line 29: "noises" -> "noise"?

Yes. Thank you. It is corrected on track-changed page 12.

Page 11, line 17: "mildly turbulent-like"
Can you express this in a more objective way. Looking at the Figures I'm not sure, what exactly is meant here. I also wonder, whether a conclusion like that can be drawn based on 3 cases only?

This expression would raise concern strictly speaking since it is descriptive and we did not go deeper. But after looking at many cases in the brighter PMC environment we considered this a likely true statement, much more than it was perceived based on what is shown here. I believe this is worth mentioning and being brought to attention. In the dimmer environment the waves are more distinctly clear than those in the brighter background.

Page 12, line 7: "three-colored" -> "three colored"

It is reworded on the track-changed page 13.

Page 12, line 12: "it plays a minor importance." -> "it is of minor importance." or "it plays a minor role."?

That sentence is reworded and please see the track-changed page 13.

Page 12, line 29: "wave lengths" -> "wavelengths"

All the spaces between the two words are deleted in the paper.

Page 13, line 21: Suggest writing "fractal perimeter dimension" rather than just "fractal dimension", because there are different fractal dimensions.

It is corrected on the track-changed page 14.

Page 15, line 13: "wave length" -> "wavelength"

It is corrected as is mentioned above.

Page 15, line 29: spaces before "Randall" and "von Savigny" missing

It is corrected.

Page 16, line 6: delete extra space after last author

It is corrected.

Page 16, line 8: add space before paper title.

It is corrected.

Figure 1, caption, line 2: add space in "and11.3"

It is corrected.

Figure 2, caption, line 5: "x5 km" -> "x 5 km"?

It is corrected.

Figure 3, ordinate label: "num of detections / bin" It's not clear (at least to me) what "bin" refers to here.

$1.0 \times 10^{-12} sr^{-2}$ is the bin size used as is stated in the caption.

Figure 4, caption, line 3: "The dashed thick black lines roughly follow"

There only seems to be a single thick black line, not multiple lines.

There are dashed magenta lines, just being pale to prevent them from blocking the view of the rainbow colored region.

Figure 6: It would be good to determine and present uncertainty estimates for the spectral slopes.

The uncertainty ranges were included as the vertical lines. They are just small. They are pretty obvious for the red lines.

**Reply-Fig-1**

[Figure]

Mexican hat power spectrum is spatially fuzzier but spectrally smoother than the Morlet counterparts. We consider Morlet the best mother wavelet for detecting gravity waves that are localized yet quasi-periodic.

Spatially averaged power spectra

**Reply-Fig-2 (left panel: the Fig.6. Right panel: Mexican hat version.)**

[Figure]

**Reply-Fig-3**

[Figure]

For FFT each orientation of basis has a corresponding spectrum. In our case it combined all in one operation.

**Reply-Fig-4**

[Figure]

Reply-Fig-4: The largest power achieved in our calculations over 2007 and 2010 summer seasons. The original albedo map (top) and the corresponding power map (bottom). This wave packet is quite typical straight waves. Large albedo power usually correspond to pretty well organized wave structures although we have said that theoretically speaking phase coherency needs to be obtained to determine their nature.

[revised manuscript text omitted]

---

## Author Comment (AC2) · 16 Sep 2017

The track-changed paper is included in this reply.

^^^^^^^^^^^^^^^^^^^^^^^^^^^^^^^^^^^^^^^^^^^^^

MINOR COMMENTS:

(1) The introduction is too much focused on the very short horizontal wavelength gravity waves and their effect on PMCs. Only in Sect.5 and in the conclusions it is mentioned that also gravity waves with quite long horizontal wavelengths (of several hundred km and more) modulate PMCs, and that their wave power can be even stronger than that of the short scales addressed in the current paper. This information, however, should already be given in the introduction.

We agree with the reviewer that the roughly -3.0 power law seems to apply to both large and small scales and it appears unnecessary to particularly focus on the small scales (20-60km in this paper). We tried systematically including the larger scales up to ~150km and found that spatial power distribution becomes flat, broad without details. This is because in CIPS the most readily observed wave features are <100km. As a result, in our scale range (20-60km) the albedo wave features from visual inspection and the albedo power show an excellent correspondence. Remember 400km is a fairly small domain to effectively include many repeats of the larger scale waves.

Therefore after p2, line 11 you should add the following information: It is also known that long horizontal scale gravity waves with wavelength of hundreds of km and longer exist in the mesosphere / lower thermosphere region (for example, Ern et al., 2011; Wachter et al. 2015). Corresponding variations are found in PMC brightness (for example, Carbary et al., 2000), and it was argued by Chandran et al. (2012) that observed PMC patterns may be caused by a superposition of large scale (>300km) and small scale (<300km) gravity waves with the large scale long period gravity waves producing the most significant increases in albedo.

We thank the reviewer for suggesting these references and we cited all of them in the track-changed text. In Ern et al. [2011] and Wachter et al. [2015] spatio-temporal analysis is applied. This is an important technique and we should make it clear that what we did for CIPS is no such analysis because CIPS has much better spatial coverage but does not have good time series. In addition, The Carbary [2000] showed nice cascading spectra from large to small scales although they did not see the small scales that most other researchers see.

I believe Chandran et al. [2012] was cited in the older version of the paper.

(2) p2 l28+29 and p3 l10 Here you are talking of "larger scale features". Please be more specific. Could this be larger scale gravity waves, or are there other effects?

The large scale features mentioned here are in a general sense, and are not meant to refer to gravity wave features specifically. We inserted a sentence to clarify this on the track-changed page 2. Also a reference by Merkel et al. [2009] is added.

(3) p3 l13+14: Here you mention that in the current paper no horizontal filtering is applied like in Chandran et al. (2010) where polynomial smoothing is used as a background, and only remaining fluctuations are analyzed. Still, your analysis technique focuses on a (narrow) range of horizontal wavelengths. This means that indirectly high pass filtering is applied by limiting the wavelet to the desired range of 20-60km (see p4l10). The main difference to Chandran et al. (2010) is that you are using a sharp cutoff, and the spectral characteristics of the remaining wave spectrum will not be altered by the filter characteristics. I think this is an important point that should be mentioned more clearly.

The reviewer is right that this work did not use preprocessing by doing the polynomial filtering. True, the polynomial filtering may alter the natural spectra. Especially when it is applied to 2D it would be harder.

Chandran et al. [2010] does this to highlight the small scales and we avoid doing it fearing that it would negatively impact the small scales. Both sides would claim that the aim is to better reveal the smaller scales gravity waves.

Chandran et al. [2010] work and ours differ in two ways. (1) They did preprocessing but we did not. (2) We focused on 20-60km but they did not. However, they did cross-track which is why their power peak occurs at around 200-250km. Zhao et al. [2015] paper obtained a peak at longer scales because they did along-track which included even larger scales.

We focused on a narrow scale range but we did not cut off other scales artificially. We demonstrated the whole albedo map but focused on the albedo wave power in a specific range.

(4) p5 l13 Please be more specific. What is the angular step/resolution used? Later in the conclusions, I learned that the step-width is 3deg. However, this information should be given also on p5 l13.

The angular resolution is 3° and it was inconsistent between Pages 10 and 14. Please look at the track-changed page 6 for a more detailed description of the resampling.

 (5) Fig.4a; Sect.4.3, p7 bottom

Since the resampling of wave events is an important step in your analysis, this should be briefly illustrated by an example. Suggestion:

If in Fig.3 an event would occur in the histogram (black curve) with 50% of the values at weaker CWT power, this event would obtain a value of freq25=0.5, and we find this event in Fig.4a at the coordinates (0.5, albedo power of the event).

It is true that Fig. 4a and the black curve in Fig. 3 are for the same set of detections. If we collapse all the data points toward the vertical axis of Fig 4a we will yield the Fig. 3 black curve since the black curve is for the set with $freq_{25}>0.2$.

However, the black curve in Fig. 3 cannot tell what exact $freq_{25}$ we have, even if we know statistically brighter clouds have higher albedo power, because there is a large scatter (STD) and they are not in a one-to-one relationship.

(6) p10, l16 and p23, caption of Fig.6: The normalization procedure was still not fully clear to me. Could you elaborate a bit more on this? Is one of the three brightness levels taken as a reference, and the others are normalized to match the reference? Is this normalization derived for a given wavelength position? How are the error bars in Fig.6 obtained?

Yes, the brightest level is taken as the standard and the normalization is toward the brightest for each category. The corresponding text is revised on the track-changed page 11.

The bar is non-zero for six of the seven points except for the middle one, or the fourth from either side. This is to make it symmetric for the double-logarithm plot. Matching the first and the 7$^{th}$ spectral data points would lead to the same results since the collapse between the different brightness levels are pretty good.

TECHNICAL COMMENTS:

(1) p1 l10 Gravity wave display morphology and clarity level -> Gravity wave display, morphology, and clarity level

It is reworded on the track-changed pages 1 and 14.

(2) p21, Fig.4 panels (a) and (b) overlap, please allow for more space between them

The two panels are further away now.

(3) p21, caption of Fig.4, about Fig.4b Please mention that the red dashed curve represents a Gaussian fit (normal distribution) to the data (black dots), while the thin black curve is a polygon that connects the data points. The magenta crosses represent normalized integrals over the number of detections below the respective analytic curves in Fig.4a.

The caption now is revised and made more clarified.

Still, the following was still unclear to me: What is the magenta line in Fig.4b? Is this a fit through the magenta crosses?
They are the same thing, just using both line and symbols. Please look at the revised figure caption.

[revised manuscript text omitted]

---

## Author Comment (AC3) · 16 Sep 2017

The track-changed paper and the reply figures are included in this pdf file.

^^^^^^^^^^^^^^^^^^^^^^^^^^^^^^^^^^^^^^^^^^^^^^^

General Comments:
This paper presents an analysis of data from the Cloud Imaging and Particle Size (CIPS) instrument on the Aeronomy of Ice in the Mesosphere (AIM) satellite that focuses on the potential to extract gravity wave information from the CIPS measurements. CIPS measures backscattered radiance at 265 nm, which provides information about the background atmosphere (excluding PMCs) at approximately 50 km altitude. An algorithm is presented to identify gravity waves with horizontal scales between _20- 60 km. Power spectra of the identified waves are determined, as well as the importance of the background albedo brightness.

First of all, we thank the reviewer to take time to read this paper. To make it clear, this work is about tracking gravity waves that reside in the PMCs rather than the CIPS 50km altitude product. The Randall et al. [2017] paper is in press and discussed the potential of the CIPS 50km product being used to conduct the gravity wave study at lower altitudes. The text is revised in the track-changed pages 1 and 4 (data section) to make this more obvious. Since the CIPS team has released the 50-55km product we should make it clear which product we have used in this study. And Randall et al. [2017] is cited.

Regarding the background brightness, it is to examine how the PMC gravity wave manifestation morphology and display clarity varies with the cloud brightness, and furthermore a universal law that controls the display clarity is extracted.

The approach to identifying and characterizing gravity waves presented in this paper is successful in some situations. However, the ability to distinguish waves (and thus study their morphology) is clearly dependent on the background albedo intensity. The presence of multiple waves in any scene, with different scale lengths and orientations, is also a challenge for classification. One suggestion would be to investigate the use of image processing algorithms to sharpen the contrast of any wave-like features before applying the detection algorithm.

We took advantage of the fact that polar summer mesopause region has a wide spread semi-organized or quasi-periodic gravity wave features, which makes it a viable means to attack the problem via only quantifying their albedo wave power, sorting these values, and extract a universal law. This law controls how waves manifest in terms of morphology and display clarity/sharpness.

This work is not meant to identify a specific wave event that bears a certain wavelength or morphology (e.g., concentric waves). We inserted a paragraph at the end of the conclusions to propose this as a future work plan. Theoretically, phase determination or wave structure coherency analysis is the ultimate means to characterize the waves more specifically. This would be a challenging task though considering the complexity of the PMC gravity wave structures.

Given the argument above saying that phase determination is a vital role, I'd say a high PMC albedo power value in CIPS almost always corresponds to a strong wave event. Please look at Reply-Fig-1_rev3 which shows the albedo map and the power maps for the maximum albedo wave power case over the two years 2007 and 2010 and it happens to correspond to a set of very clear straight waves.

P.S. We did not use any image processing/filtering (i.e., polynomial) prior to applying the algorithm because a 2D high-pass filtering will inevitably affect the accuracy of the small scales. Besides, it does not really help this current research. For example, when we look at the Fig. 11 we will see that the original albedo map reflects all scales of waves sufficiently well.

It is puzzling that the recent paper on CIPS gravity wave observations by Randall et al. [2017] (Geophys. Res. Lett. 44, 7044-7052) is not addressed in this manuscript. Four of the co-authors on this manuscript are also authors on the Randall et al. paper. Since the Rayleigh Albedo Anomaly (RAA) data used in the Randall et al. paper are also available at the AIM web site, I think it is essential that this manuscript discusses how their approach compares with the RAA method.

Randall et al. [2017] is now cited to make it clear that we did not use the 50-55km product.

Specific Comments:

1. p. 4, lines 21-22: It would be helpful to note that the Y-axis scale for the red curve in Figure 1(a) is a factor of 3 larger than the scale for the individual series.

Figure 1 caption is revised as suggested.

2. p. 4, line 31: If the individual component time series each have a maximum amplitude of ±1 unit, how can the reconstructed time series shown in Figure 1(b) have an amplitude greater than -7 at grid = 55?

The reconstruction has used wavelet transform coefficients and the corresponding basis vectors. Since continuous wavelet basis vectors are not orthogonal to each other, strictly speaking, the series cannot be recovered precisely but their variability is recovered pretty well. The reconstruction is performed to verify that wavelet transform spectra reflect the distribution of the original components in FFT space.

Please see track-changed page 5 for a slight revision.

3. p. 5, line 13: What is the angular step used to select "all radial directions"?

Sorry about this. It was not clarified well. Please look at track-changed page 6 for detailed description on the resampling. In the angular direction it is 3° increment.

4. p. 5, lines 31-33: You can't call concentric waves "characteristic" in the first sentence and "extremely rare" in the next sentence. These terms have very different meanings.

We took the reviewer's advice and it is reworded on the track-changed page 6 as "possessing a unique morphology".

What it meant was that concentric waves are by themselves characteristic features. If what it meant was "PMC waves are characterized by the concentric wave patterns" then it would conflict with the argument of "being rare".

5. p. 6, lines 23-25: Are the straight wave patterns present in the lower right quadrant of Figure 2(d)? It would be good to see more examples of straight waves, which are presumably much more common.

True. In the lower right quadrant of Figure 2(d) are straight waves. There are many more examples in Figures 7-9 and 11 that are straight waves. Please also see the Reply-Fig-1_rev3 (mentioned above) for another demonstration of the straight waves.

6. p. 7, lines 17-19: This exponential behavior is also seen in g-distribution plots of PMC brightness and ice water content (e.g. DeLand and Thomas [2015]).

DeLand and Thomas [2015] paper should be cited and please look at the track-changed page 8 for insertion.

This is not surprising since albedo power and the mean albedo is statistically in a linear relationship. PMC brightness and IWC usually can serve to describe the PMC intensity. As I look at it though, in the DeLand and Thomas [2015] the higher latitude cases are notably deviated from a straight line fitting.

7. p. 7, lines 21-23: Interannual variability of the g-distribution slope of SBUV PMC ice water content is discussed by DeLand and Thomas [2015].

Yes. It has been reviewed on track-changed page 8 as well.

8. p. 7, lines 29-31: What is the term "laminated" intended to mean here?

It is reworded as "is collapsed into". Such a word is used to be more descriptive about the fact that one bin is a small domain to contain over 65% of clouds.

9. p. 9, line 1: Why is a different analytic form needed?

Because $freq_{25}$ and mean albedo (in the elliptical region) are different variables and they hold different form of relationship with the albedo wave power although both show statistically monotonic increase with the wave power. $freq_{25}$ is used to particularly separate the brighter and dimmer cloud groups since it is a better index to characterize systematic brighter clouds within a given region such as the elliptical region we used.

10. p. 10, lines 8-10: The fits for the different power categories seem to have almost identical slopes in the left-hand panel (log scale), despite the changes in quoted slope value. Is everything correct here?

They are not the same although all three are very close to -3.0. Please look at Reply-Fig-2_rev3.

11. p. 11, lines 11-13: This is another instance where straight waves seem to be more common, which raises the question as to why there is so much initial emphasis placed on concentric waves.

If the reviewer is talking about this current study, probably Fig. 2 is what is meant for. Concentric waves are highlighted in the PMC community not because they are wide spread or is a regular occurrence but because their origin of the tropospheric storm system is unique and therefore they serve as illustrative examples to exhibit the upper and lower atmosphere connection.

In our case, we just used concentric waves to demonstrate the algorithm since these waves are well documented. We also wish to know where they fall onto the scatter plot (Fig.4a) given the fact that they are low in population.

**Reply-Fig-1_rev3**

[Figure]

Reply-Fig-1_rev3: The largest power achieved in our calculations over 2007 and 2010 summer seasons. The original albedo map (top) and the corresponding power map (bottom). This wave packet is quite typical straight waves. Large albedo power usually correspond to pretty well organized wave structures although we have said that theoretically speaking phase coherency needs to be obtained to determine their nature.

**Reply-Fig-2_rev3**

[Figure]

Reply-Fig-2_rev3: This is just to demonstrate that the three lines do have different slopes since the red and blue lines are raised in a parallel manner.

[revised manuscript text omitted]

---

## Author Response (AR2)

A note to the Editor:

Dear Editor:

The manuscript has been carefully examined and three types of changes are made, as being listed as follows:

1. Changes are made on the pages 5, 6, and 16 to address the reviewer's concerns.
2. Convoluted sentences are revised into simpler sentences without the original meanings being altered.
3. Very minor wording changes are made.

Hopefully all concerns are addressed but please let us know if any aspect is neglected.
Thank you very much ahead.

Sincerely,
Pingping

^^^^^^^^^^^^^^^^^^^^^^^^^^^^^^^^^^^^^^^^^^^^^^^^^^^^^^^^^^^^^^^^^^^^^^^^^^^^^^

**Suggestions for revision or reasons for rejection (will be published if the paper is accepted for final publication)**
Note: My comments refer to the manuscript version with highlighted changes

General comments:
In general I'm happy with the revised version of the manuscript. I still think that the results are based on some arbitrary assumptions that question the overall robustness of the results somewhat, but I have no objections to accepting the paper more or less as is.

Specific comments:

Page 5, line 1: add space in "and11.3"
The space has been added as suggested.

Page 5, line 9: "In a 6th-order Morlet wavelet (k=6) the scale s is almost precisely the period of the sinusoidal signal."

This is correct, but as pointed out in my earlier review it is important to state, whether you use the "wavelet scale" or the "Fourier equivalent scale" for further analysis. This will affect the values of the spectral exponents you get in the end.

First to answer the question directly. The scales referred to throughout the paper are the wavelet scales rather than the FFT scales. In Fig. 1 FFT components are used to generate an artificial series to illustrate that wavelet and FFT convey the same qualitative results. The text is revised on **the track-changed page 5** read as as "We emphasize here …".

In addition, I can see the reviewer's standpoint regarding the concern about "being arbitrary", and let me explain this as follows. Different transform kernels, for example, different mother wavelets (i.e., Morlet and Mexican Hat for example) deliver slightly different results quantitatively (see the previous Reply-Fig-2 from the last round) but they convey qualitatively consistent conclusions since the differences are small. What is meant here is that we are able to provide some take-home message even if not all approaches show identical results. By the way, we have chosen Morlet wavelet because it is the most optimum choice to reflect both periodicity and localization of the PMC wave signatures.

Please look at the revision on **track-changed page 16** read as "It is worth mentioning that the three α values…"

Page 6, line 11: "without having to perform systematic CWT calculations along-orbit and cross-orbit which will be two sets of calculations"

I don't understand the logic behind this statement. You don't want to do calculations along and across orbit, because you would have to do two sets of calculations, but instead you do 360 / 3 = 120 calculations? That seems like a weak argument. I suggest rephrasing this statement.

We thank the reviewer to point this out. It is agreed that this is a confusing sentence. With it **being removed**, the connection with the very next sentence sounds better (see **track-changed page 6**). To explain further, if a full 2D FFT is applied there will be more than just along-orbit and cross-orbit orientations. There is no reason to particularly address along-orbit or cross-orbit to make a point. It was said so because of the habitual thinking that along-orbit and cross-orbit are the two primary orientations that CIPS experts would care.

The overall meaning is that the current approach is a short version of the 2D FFT because the basis vectors of FFT includes all orientations.

[revised manuscript text omitted]